# What do near-optimal learning rate schedules look like?

**Hiroki Naganuma**                                                                  *hiroki11x@gmail.com*
*Mila, Université de Montréal*

**Atish Agarwala**                                                                   *thetish@google.com*
*Google DeepMind*

**Priya Kasimbeg**                                                                   *kasimbeg@google.com*
*Google DeepMind*

**George E. Dahl**                                                                   *gdahl@google.com*
*Google DeepMind*

**Reviewed on OpenReview:** *https://openreview.net/forum?id=pEsAMnmqOL*

## Abstract

A basic unanswered question in neural network training is: what is the best learning rate schedule shape for a given workload? The choice of learning rate schedule is a key factor in the success or failure of the training process, but beyond having some kind of warmup and decay, there is no consensus on what makes a good schedule shape. To answer this question, we designed a search procedure to find the best shapes within a parameterized schedule family. Our approach factors out the schedule shape from the base learning rate, which otherwise would dominate cross-schedule comparisons. We applied our search procedure to a variety of schedule families on three workloads: linear regression, image classification on CIFAR-10, and small-scale language modeling on WIKITEXT-103. We showed that our search procedure indeed generally found near-optimal schedules. We found that warmup and decay are robust features of good schedules, and that commonly used schedule families are not optimal on these workloads. Finally, we explored how the outputs of our shape search depend on other optimization hyperparameters, and found that weight decay can have a strong effect on the optimal schedule shape. To the best of our knowledge, our results represent the most comprehensive results on near-optimal schedule shapes for deep neural network training, to date.

## 1 Introduction

Modern deep neural networks are almost exclusively trained via variants of gradient descent. In these methods, the update to the parameters at each step consists of the gradient on the current batch (possibly transformed and accumulated with moving averages) multiplied by a scalar known as the *learning rate*. Setting an appropriate learning rate is essential for rapid and successful training. At best, an inappropriate learning rate will slow down the training process, and at worst it can cause it to fail entirely.

Varying the learning rate during training according to a schedule can be surprisingly beneficial. A good schedule allows models to reach much lower loss values than any constant value of the learning rate can achieve in a similar number of updates. Effective schedules in current practice consistently exhibit two high-level commonalities: an initial *warmup* stage where the learning rate increases from zero (or near-zero) to its peak value, and a later *decay* stage where the learning rate decreases back down to a much smaller value, often near zero. Analyzing the curvature of the non-linear loss surface and the resulting training dynamics can mostly explain the value of the warmup phase (Gilmer et al., 2021; Cohen et al., 2022), if not give a prescription for the exact shape of the warmup. The need to anneal noise later in training

justifies the use of some kind of final decay phase (Lee et al., 2022; Qiu et al., 2025), but also falls short of a precise characterization of the shape of the decay, or exactly when it should start. More generally there is the question of how long each phase should be, and exactly how to construct a good schedule for a given workload and number of training steps.

Despite the relative consensus on warmup and decay phases being broadly beneficial schedule components, very little is known about exactly what *shape* the learning rate schedule should take. If the schedule is even tuned at all, in many cases (for example Goyal et al. (2017); Vaswani et al. (2017); Devlin et al. (2019)) researchers will only tune a handful of schedule parameters using a fixed functional form for the warmup and decay phases: warm-up duration, peak learning rate, decay start, and final learning rate. Popular choices for the functional form of the decay are linear, inverse square root, or cosine. It is hard to imagine the optimal shape not being at least somewhat workload dependent, making guidance on how to find shapes tailored to a given workload desperately needed.

In this work, we take a first step towards characterizing how schedule shapes relate to training outcomes. Towards that end, we defined several (parameterized) learning rate schedule families (Section 3.1), including relatively flexible spline-based families that are capable of covering and extending the typical cosine, linear, and inverse-square-root curves. In Section 3.3, we develop a methodology to search these schedule families on computationally-inexpensive neural network training workloads to find near-optimal shapes within each family. This approach allows us to understand the differences between different schedule families, the benefits of different schedule shapes, and the relationship between the best-performing shapes and specific details of the workload, including other optimizer hyperparameters. In particular, this work makes the following key contributions:

- We provide the first known optimal schedule for linear regression trained using stochastic gradient descent, and use it to benchmark the efficacy of our search procedure (Section 4.1).

- We provide near-optimal schedules for the different families we defined on two neural network training workloads, a convolutional neural network (CNN) trained to classify images and a small Transformer language model workload. The best schedules from all families benefit from warmup and gradual decay, even for families such as Smooth Non-Monotonic that do not enforce these properties (Section 4.2). These results contrast with the linear regression case where the optimal schedule has no warmup and a sharp decay.

- We provide multiple lines of evidence that our search experiments were able to adequately explore all but one of our schedule families (Section 4.2).

- Finally, we show the relationship between optimal schedule shape on our workloads and other hyperparameters, specifically the AdamW $\beta_1$, $\beta_2$, and weight-decay hyperparameters, and that—in particular—weight decay strongly affects the optimal learning rate schedule shape (Section 4.3).

Our results represent an important step towards a better understanding of what makes an effective learning rate schedule for neural network training.

## 2 Related Work

From the early days of backpropagation in the 1980s and early 1990s, learning rate tuning, including schedules, has been a nuisance for anyone using neural networks. Arguably, removing the learning rate from consideration as a hyperparameter motivated the line of research on per-parameter learning rate algorithms that eventually culminated with the overwhelmingly popular Adam algorithm (Kingma & Ba, 2014). Unfortunately, despite the popularity of Adam, it does not obviate the need to tune its global learning rate hyperparameter, and the best results generally require a non-constant schedule.

It is important to distinguish the overall *scale* (the peak or "base" learning rate) from the schedule *shape* (the normalized function mapping step number to relative step size). Like any hyperparameters, base learning rates and shape parameters can in principle be tuned using random search (Bergstra & Bengio,

2012; Bousquet et al., 2017) or Bayesian optimization (Mockus et al., 1978; Snoek et al., 2012). In practice, the vast majority of the time, researchers apply a tuned base learning rate to a fixed warmup-and-decay template, at most tuning the duration of the schedule phases or trying a handful of functional forms. One representative example is Shallue et al. (2019), who predominantly used a linear decay schedule followed by a constant phase, tuning the transition point, the base learning rate, and the final learning rate, but also reported trying cosine, exponential, and other shapes. Some papers have argued for particular shapes, even including periodic ones (Loshchilov & Hutter, 2016; Smith, 2017), but they tend to be the exception. More commonly, researchers assume a particular shape and study the base learning rate more carefully; for example, Goyal et al. (2017) and Shallue et al. (2019) both look at how the base learning rate depends on batch size without investigating any relationship between the optimal schedule shape and the batch size.

When it comes to the schedule *shape* specifically, there are a handful of works that attempt to discover well-performing shapes, either offline or during the process of training a particular model. AutoLRS (Jin et al., 2021) dynamically sets learning rates via Bayesian optimization with a built-in time series model to forecast near-term training progress. Maclaurin et al. (2015) proposed gradient-based hyperparameter optimization, but their method faces meta-optimization stability and computational cost challenges, and requires co-design with the particular optimizer update rule. Hypergradient Descent (Baydin et al., 2017) adaptively updates the learning rate online using gradients, while Population-Based Training (PBT) (Jaderberg et al., 2017) evolves hyperparameters across a population of training instances, uncovering retrospectively effective schedules. Techniques that discover schedules online during training might find schedules that only perform well on a particular training trajectory, instead of reusable schedules that perform well on average over many different initial random seeds or other sources of training pipeline variation. Our work is also related to the broader *learning-to-optimize* literature, which meta-learns optimizer update rules (Andrychowicz et al., 2016; Wichrowska et al., 2017; Metz et al., 2019) and only implicitly produces a learning rate trajectory; in contrast, we directly characterize near-optimal schedule shapes within parameterized families.

## 3 Methods

Table 1: Learning Rate Schedule (Shape) Families used in our experiments

| Schedule Family | Short Name | Description |
|---|---|---|
| Constant | CON | Warmup to base LR, followed by a constant LR. |
| Cosine | COS-STD | Warmup to base LR, followed by cosine decay with a fixed exponent = 1. |
| Generalized Cosine | COS-GEN | Warmup to base LR, followed by cosine decay with a parameterized exponent. |
| Square-root Decay | SQRT | Warmup to base LR, followed by inverse square root decay. |
| Generalized Rex | REX | Warmup to base LR, followed by decay parameterized through REX (Chen et al., 2022). |
| Two-Point Spline | TPS | Warmup to base LR, followed by decay parameterized through two spline interpolation points. |
| Two-Point Linear | TPL | Warmup to base LR, followed by decay parameterized through two linear interpolation points. |
| Smooth Non-Monotonic | SNM | Parameterized by initial LR, step of peak LR, two spline interpolation points, and final LR. |

For our purposes, a learning rate schedule is a function $s(t)$ where $s$ is the learning rate at step $t$. Our goal is to find schedules that are empirically *near-optimal* within some parameterized family of schedule functions, or, in other words, as close to the true optimum $s^*$ as possible while still being within the schedule family.

In practice, the learning rate schedule is often defined as $s(t) = \alpha \cdot \phi(t/T)$ where $\alpha$ is the *base learning rate*, $T$ is the training horizon (total number of steps), and $\phi$ is a schedule *shape*[1], a function from $[0, 1]$ to $[0, 1]$. Note that there are some schedules which are defined without a horizon; we will not consider those here.

### 3.1 Learning Rate Schedule Families

To constrain the space of learning rates schedules, we defined various *learning rate schedule (shape) families* (Table 1). These are parameterized families of functions from $[0, 1]$ to $[0, 1]$ (see Appendix B.2 for details). Our Constant (CON) schedule family represents shapes without any learning rate decay (but potentially with linear warmup). The Cosine (COS-STD) schedule family represents the extremely popular decay shape

---

[1]We will sometimes use the term "relative schedule" to refer to the schedule shape in contrast to the actual, or "absolute" schedule.

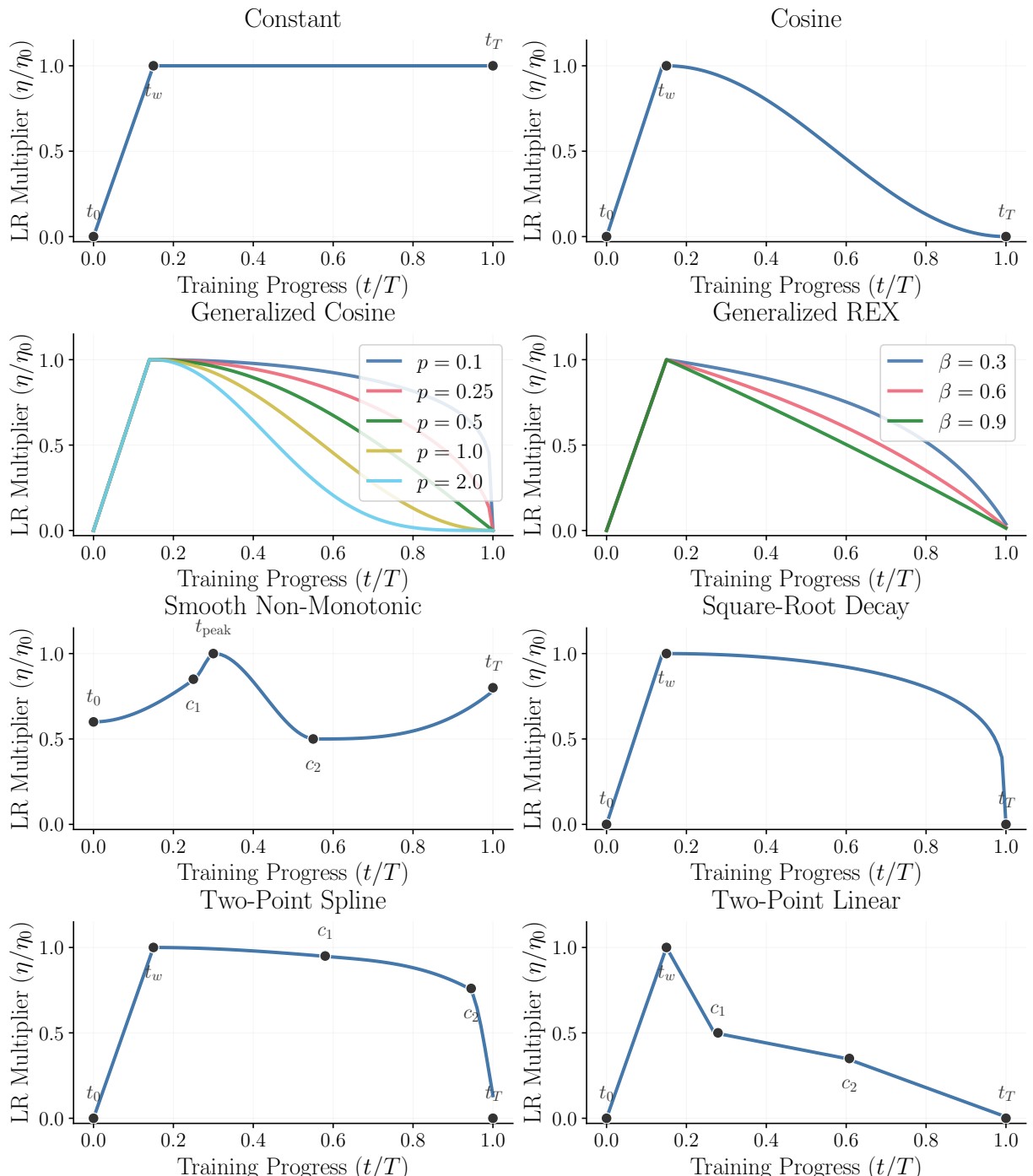

Figure 1: Learning rate schedule families used in our experiments: CONSTANT, COSINE, GENERALIZED COSINE, GENERALIZED REX, SMOOTH NON-MONOTONIC, SQUARE-ROOT DECAY, TWO-POINT SPLINE, and TWO-POINT LINEAR. Markers identify key points such as initial learning rate, warmup completion, intermediate control points, and end of training. Numerical annotations specify parameters particular to each schedule. Peak of SMOOTH NON-MONOTONIC can occur in any order compared to control points unlike other schedules.

given by $\phi(f) = (\cos(\pi f) + 1)/2$. To allow for a larger variety of shapes, we also considered the GENERALIZED COSINE (COS-GEN) family which adds a tunable power $p$ so that $\phi(f) = [(\cos(\pi f) + 1)/2]^p$. We

also studied some additional monotonically decaying schedules including the Square-root Decay (sqrt) and Generalized Rex (rex) schedules, which have been studied in Chen et al. (2022). The most flexible monotonic decay schedules we considered were Two-Point Spline (tps) and Two-Point Linear (tpl), two new families with a decay profile defined using spline interpolation with two control points (not counting the initial peak and the endpoint).

All schedule families mentioned so far also included a linear warmup, since warmup can be useful on many training workloads across many different optimizers, especially when accessing higher peak learning rates is beneficial. For completeness, we also included the Smooth Non-Monotonic (snm) schedule family, which *can* express warmup or decay, but does *not guarantee it.* This schedule family is a completely general two-control-point spline (once again not counting the peak or endpoints) that starts and ends at tunable, non-zero learning rates and has a peak at an arbitrary horizontal position independent of the control points.

### 3.2 Workloads and Experimental Setup

**Workloads**  We evaluated learning rate schedules on three different workloads, where each workload corresponds to a machine learning task defined by a dataset, model, and training objective:

- Linear Regression: minimize mean squared error (MSE) on a linear regression problem with prescribed covariance. See Appendix A.1 for details.

- Image Classification: Train a small convolutional neural network (CNN) on the CIFAR-10 dataset (Krizhevsky, 2009).

- Small Transformer-based Language Model: Train an 8 million parameter Transformer model (Vaswani et al., 2017) on WikiText-103 (Merity et al., 2016).

Details of the model architectures for the CNN and Transformer are in Appendix B.1. In order to enable high experimental throughput, we purposely selected small, computationally-inexpensive workloads where multiple training runs could fit on a single accelerator.

**Optimization-limited Regime and Training Setup**  Given a sufficiently large step budget, most (appropriately) bounded and reasonable schedules will reach a similar minimal loss, erasing the empirical differences in schedule shape performance that we are interested in studying. Even at smaller—but still large—numbers of training steps, our ability to *detect* clear differences between how different schedule shapes performed is limited. Furthermore, a natural goal of studying learning rate schedules is to train *faster* by requiring *fewer* steps to reach the same loss, making it imperative to study regimes where the number of training steps is not quite enough to get the best results. To highlight more relevant effects and to reduce these measurement issues, we purposefully selected step budgets that afford enough time to make progress on the training objective, but not enough time that most runs saturate. Training in this *optimization-limited* regime allows us to distinguish between effective schedules, which achieve better training metrics, and ineffective schedules, which produce worse training metrics. The total number of training steps and batch sizes were:

- Linear regression: 1000 steps, batch size 32.

- CIFAR-10: 1000 steps, batch size 256.

- WikiText-103: 1600 steps, batch size 512.

These training step budgets, informed by preliminary experiments, ensure that constant schedules performed somewhat poorly, allowing other schedules to demonstrate clear improvements. For linear regression experiments, we used SGD. For CIFAR-10 and WikiText-103 experiments, unless stated otherwise, we used AdamW (Kingma & Ba, 2014) with standard beta values as $\beta_1 = 0.9$ and $\beta_2 = 0.999$ and weight decay $\lambda_{WD} = 0$. Please see Appendix B for the details of other hyperparameter settings.

### 3.3 Search Procedure

At a high level, in order to answer questions about near-optimal schedules, we need a procedure to find the best (or one of the best) schedule shapes within any one of our schedule shape families. Our search procedure tries to find the schedule shape with the best (lowest) score, which we define as follows. Let $L_{\text{train}}^{(r)}(\theta, \alpha, t)$ be the training loss after $t$ steps of training with parameterized shape $\phi_\theta(t/T)$, base learning rate $\alpha$, where $r$ represents sources of randomness in the training process (e.g. weight initializations or data orderings). We define the optimal training loss by:

$$\mathcal{J}(\theta, \alpha) := \underset{r \sim \mathcal{R}}{\text{median}} \left[ \min_{0 \leq t \leq T} L_{\text{train}}^{(r)}(\theta, \alpha, t) \right] \tag{1}$$

where the median is taken over the distribution $\mathcal{R}$. By using a median, our search will prefer schedule shapes that are likely to perform well across multiple training runs with different initial weights, and tend to avoid any schedules that are hyper-specific to a particular starting point or data ordering. The optimal shape is the shape $\phi_{\theta^\star}(t/T)$ whose parameters are given by minimizing over all $\theta$ and $\alpha$:

$$\theta^\star := \arg\min_\theta [\min_\alpha \mathcal{J}(\theta, \alpha)]. \tag{2}$$

We approximate $\theta^\star$ using the two step procedure described below.

**Search step.** Our search decouples the search over the schedule parameters from the search over the optimal base learning rate $\alpha$ for a particular shape. The parameters for each learning rate schedule family were randomly sampled according to distributions detailed in Appendix B.3. For each parameter setting, we swept over 16 base learning rates from a logarithmically-spaced grid from $10^{-4}$ to $10^{-1}$. For the CIFAR-10 workload, we generated 3600 shapes for each family except the SMOOTH NON-MONOTONIC schedule family where, in some experiments, we generated an additional 36000 shapes. For the WIKITEXT-103 workload, computational constraints limited exploration to around 600 shapes per family. To score a particular schedule during the search, we used 10 PRNG seeds per schedule on CIFAR-10, and 5 different seeds per schedule for the initial search on WIKITEXT-103. Each shape was evaluated with 16 base learning rates. In total, the searches in our experiments involved over $10^7$ individual training runs for CIFAR-10, and over $10^6$ for WIKITEXT-103—large numbers enabled by our choice of small, inexpensive workloads.

**Evaluation step.** After the search, we performed an additional evaluation round on promising schedules to better rank them. We first took the top $k$ schedules ranked by median score from the search ($k = 100$ for CIFAR-10, and $k = 50$ for WIKITEXT-103), and re-trained them with 100 seeds (all pairwise combinations of 10 unique initializations and 10 unique data orderings). We again took the median for final scores, and used the DKW method (Dvoretzky et al., 1956) to compute confidence intervals when we needed to.

This two part search and evaluation strategy allowed us to focus computational resources on the most promising schedules. More details, including characterization of the noise in the procedure can be found in Appendix B.

## 4 Results

### 4.1 Linear regression: test case with ground truth

As a test of our experimental protocol, we first evaluated our search methodology on a synthetic linear regression problem with MSE loss. In this setting (described in detail in Appendix A) we define the number of datapoints $D$, the number of training steps $T$, and the data covariance spectrum $\mathbf{S}$, a $D$ dimensional vector of non-negative numbers. We chose $\mathbf{S}_k = \frac{2k}{(D+1)}$, which corresponds in the limit of large $D$ to a uniform spectrum $U(0, 1)$, normalized so that $\langle \mathbf{S}_k^2 \rangle = 1$ for convenience (the normalization just affects the base learning rate of the optimal schedule).

One advantage of this workload is that the final training loss, averaged over random initializations, can be computed as a relatively simple function of the learning rate schedule in the limit of large $D$ (Lee et al.,

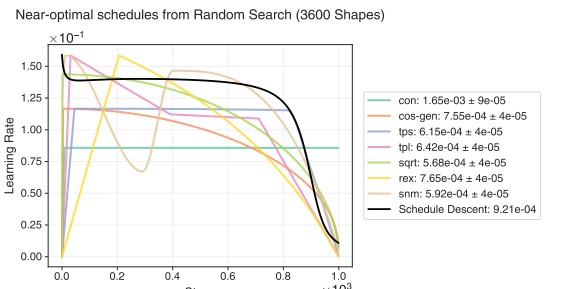
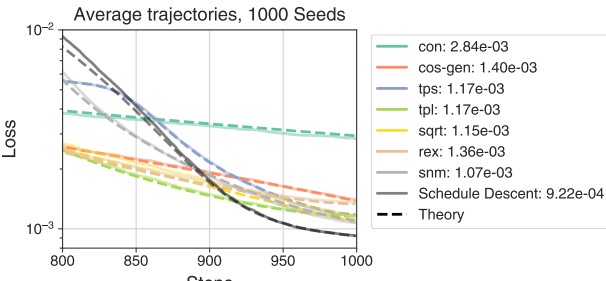

Figure 2: For a linear regression workload, schedules found via random search capture some of the features of the theoretically optimal schedule but fail to match it completely (left). Average losses appear to be better than theoretically optimal; however, when re-evaluating with 1000 seeds, searched schedules match theoretical prediction and are slightly worse than optimal (right).

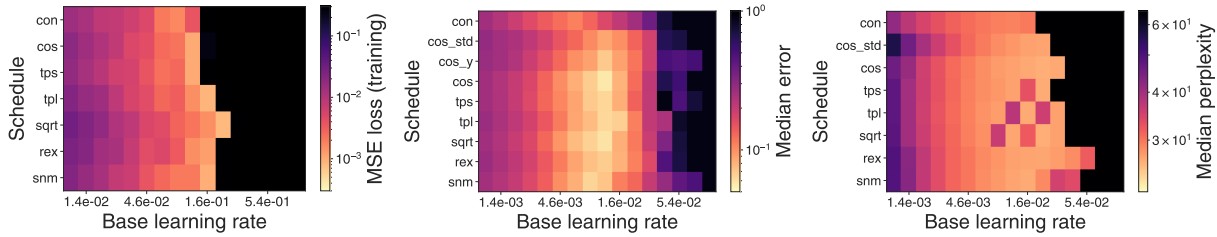

Figure 3: Training metrics versus base learning rate for best schedules in each family for linear regression (left), CIFAR-10 (middle), and WIKITEXT-103 (right). Each row corresponds to a schedule family, each column to a learning rate, and lighter colors correspond to better performance. Base learning rate is far more important for success than schedule identity, with the exception of the CONSTANT schedule which performs worse in all cases.

2022; Agarwala & Pennington, 2024). By optimizing this function, we can find the ground truth optimal learning rate schedule which minimizes the average loss at the end of training. We describe a numerical procedure for finding the optimal schedule in Appendix A.3. The result is an optimal schedule with decay and no warmup (Figure 2, left, black). The schedule is remarkably smooth given that we directly optimized the 1000 individual learning rates at each step with no constraints beyond minimizing the final training loss.

We carried out the search procedure across all our schedule families using random search with 3600 schedule shapes, evaluated with 100 seeds. The resulting schedules show some similarities in shape to the ground truth optimal schedule, with very little warmup and decay to near zero (Figure 2, left). This is especially remarkable for the SMOOTH NON-MONOTONIC schedule which has no built-in decay. The more flexible schedule families give better average final loses than the CONSTANT family, and at first glance seem to have better metrics than the theoretically predicted optimal schedule (average and standard error reported in the figure).

However, re-evaluating with more seeds shows that the true average performances are worse than the initial selection experiments suggested, and that indeed the theoretically optimal learning rate schedule is better than all the others tested (Figure 2, right). The relative ordering between the families remains similar, but the absolute performance is slightly worse than expected. We hypothesize that this is due to a combination of selection bias (we sampled the best of 3600 schedules) and the fact that this linear regression problem is known to have a log-normal distribution of the final loss (Agarwala & Pennington, 2024), making characterization of the deviations more difficult.

We found that the base learning rate is the most important predictor of the success of an (absolute, not relative) schedule (Figure 3, left). One interesting feature of the linear regression workload is that best performance occurs very close to the edge of stability. We analyze this phenomenon further in Appendix A.4;

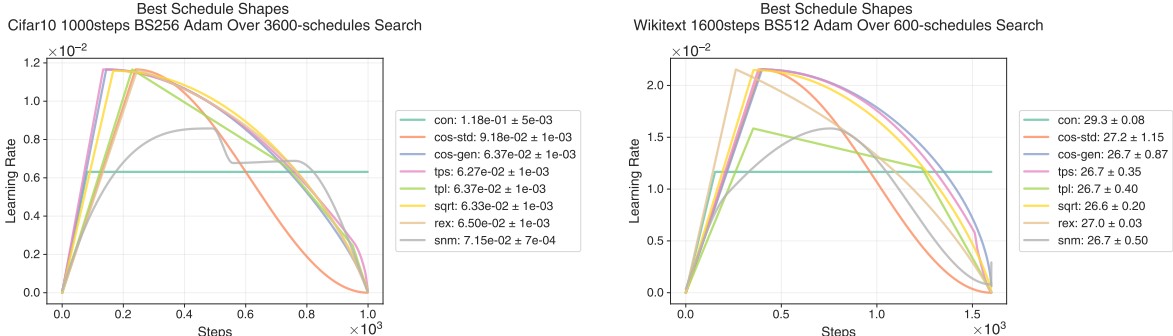

Figure 4: Near optimal learning rate schedules for CIFAR-10 (left) and WIKITEXT-103 (right). The curves represent the best absolute learning rate schedules for each family found in our search procedure which minimized final train error (CIFAR-10) or train perplexity (WIKITEXT-103). All curves show similar warmup and decay patterns in each workload, including SMOOTH NON-MONOTONIC family which is not guaranteed to have those properties. More flexible families perform better than CONSTANT and COSINE.

by studying the learning curves we can see that the optimal schedules use a large learning rate at early times that gives short term instability in large eigendirections, but help make progress in small eigendirections; the learning rate decay at the end of training is used to converge the large eigenmodes.

We note that our random search methodology does not obtain the optimal schedule in each family; this is most evident for SMOOTH NON-MONOTONIC. We used a numerical optimization scheme to find the SMOOTH NON-MONOTONIC family member closest to the ground truth optimal schedule, and found that it had a much more similar shape and was worse than the optimal schedule by only $2 \cdot 10^{-5}$, much less than the $8.5 \cdot 10^{-5}$ gap from random search (see Appendix D and Figure 14). As we will show later, the difficulty of searching SMOOTH NON-MONOTONIC in particular is a consistent theme.

Overall, our exploration of the linear regression workload suggest that:

- Our search procedure can generate schedules that obtain loss values close to optimum.

- These shapes capture qualitative and quantitative features of the optimal shape, although may not exactly match the ground truth in every respect.

- It is important to validate the efficiency of the search methods for the more flexible schedule families.

We use these lessons to guide our analysis of the CIFAR-10 and WIKITEXT-103 workloads in the remainder of this paper.

### 4.2 Near-optimal schedules for CIFAR-10 and WikiText-103 workloads

We carried out the experimental protocol described in Section 3 on our CIFAR-10 and WIKITEXT-103 workloads, over the schedule families described in Section 3.1. This leads to near-optimal schedules which have better performance than both warmup+constant and standard cosine (Figure 4). We note that the standard error of the WIKITEXT-103 experiments tends to be larger than those of the CIFAR-10 experiments, in part because some percentage of training runs using near-optimal schedules become unstable and give very large training perplexities.

We note the following important trends:

**Base learning rate is the most important factor for a good schedule.** Much like in the linear regression case, once a schedule has both warmup and decay, the base learning rate is a much more important factor in obtaining good performance as compared to the specific family used (Figure 3, plotted for best schedules for each family). Note that unlike the linear regression case, for non-linear models it appears that

there is a non-trival range of learning rates larger than the optimal, which don't lead to divergent training. For all families except CONSTANT, extra tuning budget is generally better spent on finetuning base learning rate as opposed to more detailed tuning of schedule hyperparameters.

**Warmup and monotonic decay are both crucial.** For both workloads warmup fractions were non-trivial (from $10 - 30\%$ of total training time), and the decay took up the rest of the time. Most strikingly, this is true for the SMOOTH NON-MONOTONIC schedule, which does *not have warmup or decay built in* but nonetheless "discovers" these techniques via random search. This suggests that warmup and monotonic decay are not just the result of our limited ability to optimize schedules in expensive workloads, but a fundamental feature of good schedules in deep learning problems. Indeed, if researchers had carried out a search like ours on a family like SNM in the early days of machine learning research, *they may have discovered warmup and decay earlier.*

**Optimal schedules for deep learning workloads can differ significantly than those for linear regression workloads.** The optimal schedule for our linear regression workload has no warmup. Additionally, it has a flat, large learning rate for most of training followed by a sharp decay. This is very different from our two deep learning workloads which benefit from nontrivial warmup and favor gentler decay.

**Flexible families provide small but significant gains.** For CIFAR-10, we found that the more flexible families generally gave shapes that were significantly different than COSINE, and achieved lower training error as well: the best flexible families (TWO-POINT SPLINE, SQUARE-ROOT DECAY, GENERALIZED COSINE, TWO-POINT LINEAR) reached median training errors of 0.063–0.064, compared to 0.092 for COSINE (Figure 4). For WIKITEXT-103, the optimal schedule shapes are different from COSINE, with smaller median perplexities (26.6–26.7 vs. 27.2 for COSINE), though the larger standard errors make more precise statistical statements difficult (Figure 4).

We found similar results when selecting schedules using test metrics, with somewhat increased variation in schedule shapes and smaller base learning rates in the case of CIFAR-10 (see Appendix E and Figure 15).

To further validate our search procedure, we conducted additional checks on the near-optimality of our search procedure (Appendix F). We compared the top-3 shapes in each family, and found that within-family similarity was very high for all shapes except SMOOTH NON-MONOTONIC. We also conducted coordinate-wise linesearches on TWO-POINT SPLINE and SMOOTH NON-MONOTONIC, which suggested that TWO-POINT SPLINE is well-optimized but SMOOTH NON-MONOTONIC is not. ECDF analysis suggests that the difficulty of optimizing SMOOTH NON-MONOTONIC is due to the large parameter space rather than any fundamental limit of the schedule family, similar to the results on linear regression. Overall, all families except SMOOTH NON-MONOTONIC can be reasonably considered "near-optimal" under our search procedure.

### 4.3 Workload variations

In our workload definitions, we focused on the case where all optimizer hyperparameters are fixed except for the learning rate at each step. In practice, these hyperparameters may also need to be tuned, raising the question of how—and to what extent—the optimal schedule shape depends on the exact values of these other training hyperparameters?

To answer this question, we conducted a series of experiments where we independently varied the ADAMW hyperparameters $\beta_1$, $\beta_2$, and $\lambda_{WD}$ (the decoupled weight decay parameter). We focused on the TWO-POINT SPLINE family, which gave strong results in our previous experiments due to its balance of flexibility and searchability. For each hyperparameter, we selected a grid of values, which defined a set of experimental conditions. For each condition, we performed our search and evaluation protocols to select the best schedule for that particular hyperparameter value. In general, our search procedure will almost always return a "best" shape for a given setting of (say) $\beta_1$ that looks at least slightly different from the shape returned for another value, even if the best shape has no meaningful dependence on $\beta_1$. However, if $\beta_1$ has little effect on which shapes perform well, then a shape selected with $\beta_1 = 0.8$ should perform just as well when re-evaluated at $\beta_1 = 0.9$ as the shape originally found using $\beta_1 = 0.9$. To rule out the case where superficially different

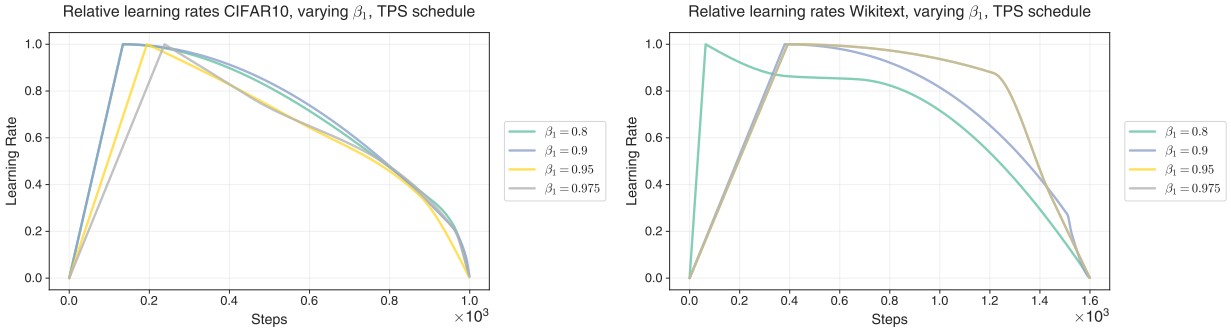

Figure 5: Optimal schedule shape varying $\beta_1$. Relative learning-rate schedules that minimise loss after tuning $\beta_1$. Left panel shows CIFAR-10; right panel shows WIKITEXT-103. Low $\beta_1$ values give a short warmup and a brief high-rate plateau, while high $\beta_1$ values extend both. Results for $\beta_2$ are presented in Appendix G.

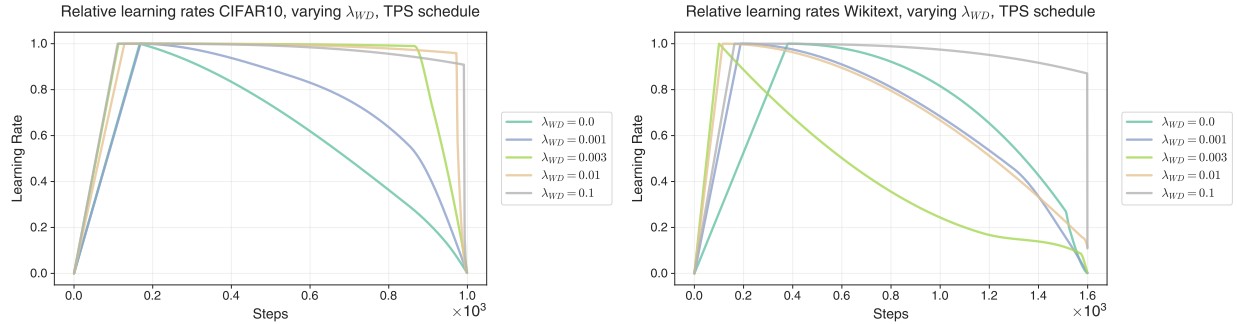

Figure 6: Optimal Shape varying Weight Decay | Relative learning rate schedules that yield the lowest loss after tuning weight decay. Left panel shows CIFAR-10; right panel shows WIKITEXT-103. Small decay values lengthen the warmup, while large decay values keep the rate high until the final training steps. Both tasks trace the same pattern.

optimal shapes do not lead to detectably different training outcomes, we re-evaluated the optimal shape found in each condition under every other condition.

All experiments were done on both the CIFAR-10 and the WIKITEXT-103 workloads. We describe the results in the remainder of this section.

**Varying AdamW** $\beta_1$**.** We found only minor optimal shape variations when varying $\beta_1$ on CIFAR-10 (Figure 5, left). On WIKITEXT-103, larger $\beta_1$ seemed to favor later decay (Figure 5, right). In order to further test the significance of results, we cross-evaluated the best schedules found for each $\beta_1$ across the other values of $\beta_1$ (Appendix H), which suggested that only $\beta_1 = 0.8$ and $\beta_1 > 0.8$ on WIKITEXT-103 showed convincing evidence of changing the optimal schedule. The effect of $\beta_2$ on optimal schedule shape appeared minimal and is presented in Appendix G with cross evaluation tables in Appendix H.

**Varying** $\lambda_{WD}$**.** Interestingly, varying the (decoupled) weight decay strength revealed a strong relationship between $\lambda_{WD}$ and the learning rate schedule. In both workloads, increasing the weight decay favored schedules that decayed later (Figure 6). Our cross-evaluation experiments generally showed that the schedules selected for a particular value of $\lambda_{WD}$ tend to be optimal for that specific value (Tables 9 and 10), as we would expect if changing the weight decay strength truly changed what schedule shapes performed well. Note that of the conditions we searched, $\lambda_{WD} = 0$ gives the best overall performance (as we would expect since we are comparing *training* metrics rather than validation or test metrics).

We also varied the training horizon on WIKITEXT-103, finding that longer horizons favor gentler decay while the warmup fraction remains stable (see Appendix I and Figure 22).

## 5 Discussion

### 5.1 Lessons about search methodology

Our search methodology was designed to be simple, but it is not particularly efficient. Crucially, the search always works better with more seeds and more search points. The choices we made in our experiments generally seemed to yield plausibly near-optimal schedules. In the case of CIFAR-10, our random search procedure seems sufficient at the small scale of our experiments; on WIKITEXT-103 we were also generally able to draw conclusions, but there was more variability—in part due to the reduced number of seeds and search points, and in part due to the increased variance in the stability of training runs with good median final train perplexities.

Our results (such as those presented in Figure 3) validated our choice to separately optimize the base learning rate for each schedule shape. By tuning the base learning rate for each shape, we isolated the effect of the shape from the strong base learning rate effects (that could appear implicitly through shapes that stay close to their peak longer). The efficiency of the random search can obviously be improved in various ways, but we have not even really optimized the allocation of computational resources across search points, seeds, and evaluation, let alone explored more sophisticated search strategies.

Although generally sufficient, for the SMOOTH NON-MONOTONIC family (which was maximally flexible but correspondingly hard to search) our search procedure wasn't quite as effective as it needed to be. This was in part because warmup and decay seem to be integral parts of any successful learning rate schedule; schedules with these properties compose a small fraction of the configuration space of the SMOOTH NON-MONOTONIC family.

Searching more flexible families (perhaps ones even more flexible than the SMOOTH NON-MONOTONIC family) could benefit from improved techniques. Random search could be augmented with better priors, or multi-round searches that incorporate feedback could be used, such as Bayesian optimization or evolutionary algorithms. These methods could also improve search efficiency in the more easily searchable families, which could be of practical interest for more expensive workloads. However, although more efficient, these kinds of techniques can make the results harder to interpret since they could potentially introduce a misleading bias towards different parts of the search space, depending on the exact details.

Though we applied our search methodology to learning rate schedules, most optimizers contain other scalar hyperparameters that could benefit from scheduling. The most obvious is momentum (Lee et al., 2022; Ferbach et al., 2025), but Adam $\beta_2$ and $\lambda_{WD}$ may benefit from scheduling as well. A natural extension of our work is to search flexible schedule families on these other parameters as well.

### 5.2 Practical takeaways

The most important, if somewhat obvious, takeaway from our work is that without optimizing the base learning rate, trying to optimize the schedule shape isn't very meaningful. Anyone attempting to search over or tune schedule shapes should make sure to devote sufficient resources to tuning the base learning rate. Also, when trying some new learning rate schedule family, the base learning rate must be tuned anew in order to minimize false negative results.

Our results confirm that, at a high level, the common practice of a warmup period followed by monotonic decay is indeed a consistently effective strategy. The best members of the SMOOTH NON-MONOTONIC family, which does not guarantee either phase, showed these features. This result suggests that near-optimal learning rate schedule searches could have revealed these rules of thumb earlier on in the history of deep learning.

Our analysis of the linear regression workload represents, to the best of our knowledge, the first optimal schedule for a non-trivial, high-dimensional problem in that class. It is interesting to note that the optimal schedule for linear regression is relatively "smooth", despite no such constraints induced by the optimization procedure; each of the 1000 learning rates at each step was individually optimized. The schedule favors no warmup as well as a relatively flat schedule with a sharp decay at the end of training. The shape of the optimal schedule in linear regression depends on the training horizon, as well as the spectrum of the data

covariance. We hypothesize that changing these choices might change the decay profile, but shouldn't make warmup suddenly become useful.

Interestingly, the optimal schedules for CIFAR-10 and WIKITEXT-103 are quite different from the optimal schedule for the linear regression workload, namely in that warmup is useful across families, workloads, and workload variations. Comparing our results on linear regression and neural net training indicate extreme caution is necessary when applying principles derived from studies of convex optimization to the non-convex, non-linear setting of optimization in deep learning, specifically when trying to understand useful principles for early and intermediate time training.

Our results on CIFAR-10 and WIKITEXT-103 suggest that, if tuning resources are available, it is indeed worth considering schedules beyond popular cosine decay schedule. The ability to change the shape of the decay gave benefits in both train and test set metrics. In our workloads, these gains tended to be small but significant.

The differences in final training metrics between the various more flexible families, and between top members of each family, seem to be small. This result suggests that it might not be worth searching *even more flexible* schedule families. Even the GENERALIZED COSINE family, with one more parameter than COSINE, captured significant gains in the CIFAR-10 case. Another popular and slightly more general cosine family than COSINE uses a non-zero final learning rate (COS-STD). When this final learning rate is tuned well enough, we saw some gains over COS-STD, but still didn't get as good results as COS-GEN (Appendix C.1). If computational resources are more abundant, then the TWO-POINT LINEAR or TWO-POINT SPLINE families likely have more than enough flexibility to capture schedules very close to optimal.

Our ability to find near-optimal schedules may be useful for research into automatic learning rate selectors. By measuring different quantities during training, we could hope to predict the (known) optimal learning rate schedule as a function of these quantities. This might relate to simple measurements like the loss trajectory, gradient norms, or simple Hessian statistics that can be measured cheaply during training. We can do even better in the linear regression setting where we can solve for the optimal schedule, and compute many of these quantities analytically.

The generality of our conclusions is limited primarily by our small number of workloads. Applying our methodology to a larger variety of workloads will give more evidence of which features of optimal schedule shapes are general versus specific. Nonetheless, even this initial study has clarified some mysteries and gives a path to better understanding of learning rate schedules.

We close this section with two clarifications. First, the paper is empirical and we do not provide formal optimality guarantees: "near-optimal" refers to the best schedule we can find within a parameterized family using a finite search budget, validated for each family in Appendix F. A theoretical analysis of near-optimality remains an open direction. Second, on the qualitative gap between linear regression (no warmup, sharp decay) and deep learning (warmup, gentle decay): in linear regression the curvature is fixed by the data covariance, so an aggressive learning rate from step one is safe, whereas progressive sharpening and curvature dynamics near the edge of stability (Cohen et al., 2022; Gilmer et al., 2021) mean that in deep learning the largest Hessian eigenvalues can grow substantially during early training, making warmup useful for letting the curvature reach a state in which the peak learning rate is tolerable. We offer this as an interpretation rather than a demonstrated mechanism.

## 6 Conclusion

This research addressed the fundamental question of identifying nearly optimal LR schedule shapes by clearly distinguishing between the schedule's scale and shape. We perform comprehensive empirical evaluations of various families of schedules and workloads. Our key findings revealed consistent structural features in effective learning rate schedules: an initial warmup phase, smooth decay after reaching peak learning rate, and very low learning rate towards the training end. Highly flexible schedule families, such as SMOOTH NON-MONOTONIC, showed significant potential to discover theoretically optimal shapes. However, practically, moderately structured families like GENERALIZED COSINE schedules—particularly those with adjustable exponent parameters—offer a better balance between performance and exploration efficiency. Additionally,

we demonstrate how factors such as optimizer choice, hyperparameters, and weight decay influence not only the optimal LR scale but also its shape, emphasizing the context-dependent nature of LR schedule design.

The systematic evaluation framework and insights presented here provide practical guidance for future research and more effective deep learning model training. Future challenges include developing more efficient schedule exploration methods, generalizing findings to larger-scale models and tasks, and designing adaptive schedules suitable for different computational budgets.

Specific extensions raised by this work include scaling our search to LLM-scale training horizons, characterizing the dependence of optimal shape on batch size, exploring horizon-free families and nonlinear warmup, and applying the same methodology to modern optimizers such as Lion (Chen et al., 2023), Muon (Jordan et al., 2024), and cautious weight decay (Chen et al., 2026).

## Acknowledgements

We thank Shankar Krishnan, Sourabh Medapati, Dougal Maclaurin and Vincent Roulet for their discussions on the methods, support in experimentation, and detailed feedback on the manuscript.

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

# Appendix

## A  Linear regression workload details

### A.1  Workload definition

The linear regression workload consists of a linear model on a $P$ dimensional parameter vector $\boldsymbol{\theta}$, optimized on $D$ data points according to the following loss function:

$$\mathcal{L}(\boldsymbol{\theta}) = \frac{1}{2D}||\mathbf{z}||^2, \ \mathbf{z} \equiv \mathbf{J}\boldsymbol{\theta} - \mathbf{y}_{tr} \tag{3}$$

Here $\mathbf{J}$ is the $D \times P$ dimensional Jacobian, and $\mathbf{y}_{tr}$ are the $D$ targets.

We train the model using SGD with learning rate schedule $\alpha_t$, randomly sampling $B$ out of the $D$ datapoints independently every step. The dynamical equations can be written as

$$\boldsymbol{\theta}_{t+1} - \boldsymbol{\theta}_t = -\frac{\alpha_t}{B}\mathbf{J}^{\mathrm{T}}\mathbf{P}_t\mathbf{z}_t. \tag{4}$$

Here $\mathbf{P}_t$ is a sequence of random, i.i.d. diagonal matrices with exactly $B$ random 1s on the diagonal, and 0s everywhere else, which represents the sampling process.

The dynamics of this model can be written in terms of the residuals $\mathbf{z}$ alone:

$$\mathbf{z}_{t+1} - \mathbf{z}_t = -\frac{\alpha_t}{B}\hat{\Theta}\mathbf{P}_t\mathbf{z}_t. \tag{5}$$

where $\hat{\Theta} = \mathbf{J}\mathbf{J}^{\mathrm{T}}$ is the empirical neural tangent kernel.

This is the dynamics we directly simulate. We initialize $\mathbf{z}_t$ as an i.i.d. random vector whose elements are drawn from $\mathcal{N}(0, 1)$. We initialize $\hat{\Theta}$ using a fixed diagonal matrix $\boldsymbol{\Lambda}$ and a uniformly sampled $D \times D$ orthogonal matrix $\mathbf{U}$:

$$\hat{\Theta} = \mathbf{U}\boldsymbol{\Lambda}\mathbf{U}^{\mathrm{T}} \tag{6}$$

We used a uniform distribution on $\boldsymbol{\Lambda}$, with eigenvalues given by $\lambda_k = 2k/(D+1)$ for $k$ from 1 to $D$. This ensured that the average eigenvalue of $\hat{\Theta}$ is 1. For all our simulations, we set $D = 500$ and trained for 1000 steps.

### A.2  Average loss curves in the high-dimensional limit

In the limit of large $D$, the loss curves of the above linear regression problem concentrate around their average (Lee et al., 2022). The average is described by tracking the dynamics of the variances in each eigenmode of $\hat{\Theta}$ (Agarwala & Pennington, 2024). More precisely, Equation 8 below is derived in the joint regime where the parameter dimension $P$ and the data dimension $D$ both grow with $P/D$ and $\beta = B/D$ held fixed; our

experiments use $P = D = 500$, finite but large enough that the approximation matches 1000-seed averages well (Section A.4). We define:

$$\mathbf{p}_t = \mathrm{diag}(\mathbf{U}^{\mathrm{T}}\mathrm{E}[\mathbf{z}_t\mathbf{z}_t^{\mathrm{T}}]\mathbf{U}) \tag{7}$$

where the expectation is taken over the initialization and sampling randomness.

In the high dimensional limit, the dynamics of $\mathbf{p}$ is described by the following $D$ dimensional linear recurrence relationship:

$$\mathbf{p}_{t+1} - \mathbf{p}_t = \left[(\mathbf{I} - \alpha_t\mathbf{\Lambda})^2 - \mathbf{I} + \frac{1}{D}(\beta^{-1} - 1)\alpha_t^2\mathbf{\Lambda}\mathbf{1}\mathbf{1}^{\mathrm{T}}\mathbf{\Lambda}\right]\mathbf{p}_t \tag{8}$$

where $\beta = B/D$. The average loss can be written as

$$\mathcal{L}(\mathbf{p}) = \frac{1}{2D}\mathbf{1}^{\mathrm{T}}\mathbf{\Lambda}\mathbf{p}. \tag{9}$$

Combined, Equations 8 and 9 imply that the loss can be written as an analytic function of the learning rate schedule. This will allow us to use standard optimization techniques to find optimal learning rate schedules for a given $\mathbf{\Lambda}$, batch size $B$, and number of steps $T$.

## A.3 Schedule descent

In general, there are no analytic solutions to the optimal learning rate schedule for non-trivial $\mathbf{\Lambda}$. There is a limit where Equation 8 can be approximated with an ODE, which gives rise to a calculus of variations problem to optimize the learning rate schedule for lowest loss after some fixed number of steps. However this problem is intractable when $\mathbf{\Lambda}$ has more than one unique eigenvalue.

Instead, we can directly optimize the schedule with gradient descent via Equations 8 and 9—this is exactly the well-defined finite-dimensional optimization problem one obtains by combining (8) and (9) without any ODE approximation, by unrolling the recurrence for $T$ steps. The resulting objective is non-convex in $\{\alpha_t\}$, and a detailed theoretical analysis of its complexity is a natural follow-up. More explicitly, $\mathbf{p}_T$ is a function of the $T$ dimensional vector $\alpha_t$, whose elements are the learning rate at each step $t$. Equation 8 implies that the loss after $T$ steps can be written as $\mathcal{L}(\{\alpha_t\})$, a loss function of the output of a neural network, whose layers are a quadratic function of $\alpha_t$. This means that the standard machinery of deep learning can be used to differentiate through $\alpha_t$ to do gradient based minimization of $\mathcal{L}(\{\alpha_t\})$.

We carried out a *schedule space gradient descent* (schedule descent for short) algorithm as follows. We initialized by finding a good constant schedule $\alpha_t = C$ using a logarithmically spaced grid search.

We then carried out standard gradient descent on the objective $\log(\mathcal{L}(\mathbf{p}_T(\{\alpha_t\})))$ with a constant learning rate schedule. We found that optimizing $\log(\mathcal{L})$ sped up convergence dramatically without having to use a (meta) learning rate schedule for the schedule descent. We also added one additional update rule to schedule descent: if the final loss was above a certain threshold (set to 10 in our setup), then instead of taking a gradient descent step, the optimizer multiplies all learning rates uniformly by 0.3. This helped us deal with the fact that the optimal schedules are often close to ones that lead to divergent training, at which point no gradient signal can be obtained from the current schedule.

We found that the schedule progression was generally smooth (Figure 7). Decay at the end quickly developed, which allowed the median base learning rate to increase as well. It only took around 300 steps for the loss to stop changing appreciably and for the schedule shape to converge. One remarkable feature is that the schedule appears to be relatively smooth, even though the 1000 discrete learning rates were simultaneously optimized with no restriction. We only enforced "smoothness" in the constant initial condition.

We used the schedule obtained from our search as the optimal schedule in our study of linear regression; as we showed in the main text, it indeed had the lowest average loss out of all schedules we studied.

## A.4 Loss curves from random search

One interesting feature of the loss curves of the schedules found using our searches in the various families is that many of the best schedules give non-monotonic loss curves (Figure 8). This suggests that many good

Figure 7: Progress of schedule descent optimizer on linear regression workload. Each curve represents the state of the schedule at different steps of the optimization of the loss at the final training point $T = 1000$. The first step finds a good constant learning rate schedule; after that the algorithm runs standard gradient descent with a fallback if the schedule leads to diverging training. Final loss, and the corresponding schedule seemed to converge after around 300 steps. The schedule is represented by a $T$ dimensional vector of discrete learning rates, continuity was not enforced beyond the initial condition.

schedules temporarily exceed the *edge of stability* — that is, they use learning rates larger than the largest convergent constant learning rate. This can be beneficial, since exceeding the edge of stability helps the optimizer make more progress on smaller eigenmodes.

As long as the larger eigenmodes are not unstable for too long, they can be converged during the learning rate decay phase where the learning rate drops below the edge of stability. Here these large eigenmodes converge more quickly than the smaller eigenmodes, so progress on the objective is not harmed by the brief foray into instability. This is similar to the fact that in the non-convex setting, there are stable high frequency oscillations in the large eigendirections for a significant portion of training (Cohen et al., 2022).

One interesting feature of the linear regression workload is that it favors no warmup. This provides another line of evidence that warmup is indeed a useful feature due to non-convexity and curvature dynamics in deep learning workloads (Gilmer et al., 2021), and can't be explained with convex reasoning alone.

### A.5 Theory and experiment comparison

Given that Equation 8 represents a high-dimensional limit of the dynamics of the linear regression problem, we can ask: how well does this reflect the average in practice? To validate our approach, we answered this question by comparing the learning trajectories of the best schedules from each family, averaged over 1000 seeds, to the theoretical predictions of the loss for different schedules (Figure 8). For all schedules, there was good agreement between the theory and the empirical averages. For schedules with smaller maximum learning rates (like the constant schedule or COS-GEN, the match was good through the whole trajectory.

However, the better schedules spent time at or even slightly above the edge of stability, and showed worse agreement with theory at intermediate times. The theory captured, for example, non-monotonicity in the loss curves, but there were quantitative differences at intermediate times. However these differences vanished at late times. We speculate that for these trajectories, there were a small number of eigendirections (corresponding to the largest few eigenvalues) which dominated these parts of the trajectory, which leads to a breakdown of some of the assumptions needed to apply the full high-dimensional version of the theory. There is indeed a more general version of Equation 9 which is exact at finite dimensions but not tractable to simulate and optimize as it contains $D^2$ coupled linear equations (Agarwala & Pennington, 2024).

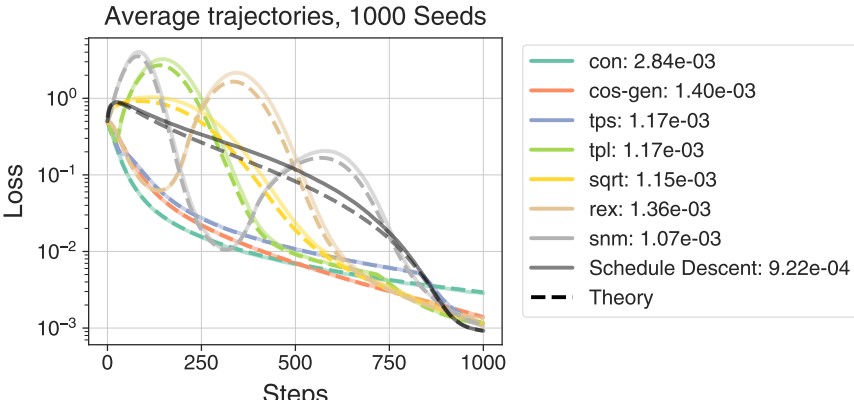

Figure 8: Training loss curves for linear regression workload. When averaged over 1000 seeds, theory predicts early time and late time behavior well, but shows some systematic deviations at intermediate times where dynamics is near/above edge of stability, and loss curves can become non-monotonic.

These discrepancies vanish at late times because the learning rate decay causes the large eigenvalues to converge to near-zero values, compared to the small eigenvalues which dominate the loss at late times (whose dynamics is well suited to the high-dimensional approximation). We believe simulating the full system (which requires analysis of a $D^2 \times D^2$ matrix) would lead to very minute differences in the optimal schedule shape, none of which affect our conclusions. We leave detailed analysis of the finite-size deviations to future work.

### A.6 Matching optimal schedule with the Smooth Non-Monotonic family

In Figure 14, we found a member of the SNM family which best-approximated our optimal learning rate schedule from schedule descent. We found this member by minimizing the MSE loss between the schedule values by optimizing the parameters of the SNM family member.

Because our SNM implementation used a Numpy based code that was not easily convertible to a differentiable JAX form, we ended up implementing an approximate GD algorithm using sampling. We performed discrete coordinate descent, using small changes in one parameter at a time to give a discrete approximation to the gradient.

It is likely that there can be small improvements made to the SMOOTH NON-MONOTONIC member in terms of minimizing its $L^2$ distance to the optimal schedule; however the point still stands that there exist members of the SMOOTH NON-MONOTONIC family which much more closely approximate the optimal schedule. Indeed it is not even clear that minimizing the $L^2$ distance to the optimal schedule is what minimizes the loss; nonetheless, this new SMOOTH NON-MONOTONIC member had significantly lower loss than the schedule found via our search procedure, and was quantitatively close to the performance of the optimal schedule.

# B  Experimental Details

In this section, we describe model architecture detail and how each schedule family parameterizes the shape of the schedules used in our study. Full implementation details are available at `https://github.com/google/init2winit/tree/master/init2winit/projects/optlrschedule`.

## B.1  Model Architecture

The CNN architecture in Table 2 targets CIFAR-10 classification. It begins with a convolutional layer of 32 filters (3x3), followed by ReLU and max pooling operations. A second convolutional layer employs 64 filters (3x3), again using ReLU and max pooling. After pooling, the output is flattened and processed by a dense layer of 256 units with ReLU activation. A final dense layer then outputs logits corresponding to the 10 CIFAR-10 classes.

Table 2: Detailed architecture of the CNN model used for CIFAR-10 classification. The model comprises two convolutional blocks, each followed by ReLU activation and max pooling, and two fully connected layers.

| Layer | Details | Output Shape |
|---|---|---|
| Input | Image ($32 \times 32 \times 3$) | (32, 32, 3) |
| Conv2D | 32 filters, kernel size = (3, 3), stride = 1 | (30, 30, 32) |
| Activation | ReLU | (30, 30, 32) |
| Max Pooling | window size = (2, 2), stride = 2 | (15, 15, 32) |
| Conv2D | 64 filters, kernel size = (3, 3), stride = 1 | (13, 13, 64) |
| Activation | ReLU | (13, 13, 64) |
| Max Pooling | window size = (2, 2), stride = 2 | (6, 6, 64) |
| Flatten | Flatten feature maps to vector | (2304) |
| Dense | Fully connected, 256 units | (256) |
| Activation | ReLU | (256) |
| Dense | Fully connected, 10 units (logits) | (10) |

Table 3: Architecture of the TRANSFORMER language model used for WIKITEXT-103 training. The model follows a standard decoder-only TRANSFORMER structure with multiple self-attention and feedforward layers.

| Layer Type | Configuration | Output Shape |
|---|---|---|
| Input Embedding | Token and position embeddings | (batch, 128, 256) |
| Transformer Block (repeated $N$ times) | | |
|    Multi-Head Self-Attention | $h$ heads, each with $d_k$, $d_v$ | (batch, 128, 256) |
|    LayerNorm + Residual | Pre-norm | (batch, 128, 256) |
|    Feedforward Network | Two linear layers with activation | (batch, 128, 256) |
|    LayerNorm + Residual | Pre-norm | (batch, 128, 256) |
| Output Projection | Linear layer to vocab size | (batch, 128, 10,000) |

Table 3 summarizes the architecture of the TRANSFORMER model applied to WIKITEXT-103. It consists of a stack of standard decoder-style TRANSFORMER blocks. Each block includes multi-head self-attention, layer normalization, and feedforward layers. Token and positional embeddings are used at the input, and final predictions are made through a linear projection to the vocabulary dimension.

## B.2  Schedule Families

Figure 1 illustrates representative parameterizations from each schedule family.

The Smooth Non-Monotonic family provides the highest degree of flexibility, parameterizing the schedule shape with seven parameters. It imposes no explicit constraints at the beginning or end of the learning rate schedule, nor does it require warmup or decay periods.

The Two-Point Spline and Two-Point Linear families, each defined by five parameters, enforce linear warmup constraints. Two-Point Spline schedules rely on spline interpolation after the warmup period, whereas Two-Point Linear employs linear interpolation.

The Generalized Cosine family explores the exponent parameter within Cosine, also incorporating a linear warmup period. Cosine fixes the exponent at 1, varying only the warmup duration. Generalized Rex explores variations of the beta parameter within the REX schedules (Chen et al., 2022). Finally, the Square-root Decay schedule applies a square-root decay to the learning rate.

### B.3 Hyperparameter Search Space

This section describes the search spaces for hyperparameters associated with each schedule family and workload.

For base learning rates, linear regression experiments used 16 evenly log-spaced values from $[0.01, 1.0]$; all other experiments used 16 evenly log-spaced values from $[0.001, 0.1]$. All experiments used AdamW as the optimizer. Unless specified otherwise, weight decay was set to 0. Section 4.3 describes the ablation study in detail.

For each schedule family, the hyperparameters and their respective search ranges are presented in Table 4.

Table 4: Hyperparameter search spaces for all schedule families.

| Schedule Family | Hyperparameter | Search Range |
|---|---|---|
| Constant | warmup_steps (fraction) | $[0, 0.25]$ |
| Cosine | warmup_steps (fraction) | $[0, 0.25]$ |
| | exponent | $[0, 2.0]$ |
| | alpha | $0.0$ |
| Square-root Decay | warmup_steps (fraction) | $[0, 0.25]$ |
| | alpha | $[0, 2.0]$ |
| Generalized Rex | warmup_steps (fraction) | $[0, 0.25]$ |
| | beta | $[10^{-8}, 32]$ |
| Two-Point Spline | x0 | $[0.01, 0.25]$ |
| | y1 | $[0.1, 1.0]$ |
| | delta_x1 | $[0.0, 1.0]$ |
| | delta_x2 | $[0.0, 1.0]$ |
| | delta_y2 | $[0.0, 1.0]$ |
| Two-Point Linear | x0 | $[0.01, 0.25]$ |
| | y1 | $[0.1, 1.0]$ |
| | delta_x1 | $[0.0, 1.0]$ |
| | delta_x2 | $[0.0, 1.0]$ |
| | delta_y2 | $[0.0, 1.0]$ |
| Smooth Non-Monotonic | y_start | $[0.0, 1.0]$ |
| | y_end | $[0.0, 1.0]$ |
| | x_peak | $[0.0, 1.0]$ |
| | y1 | $[0.0, 1.0]$ |
| | delta_x1 | $[0.0, 1.0]$ |
| | y2 | $[0.0, 1.0]$ |
| | delta_x2 | $[0.0, 1.0]$ |

### B.4 Noise characteristics of CIFAR-10 measurements

To choose the number of seeds used in our search and evaluation protocol, we conducted preliminary experiments using the TWO-POINT SPLINE schedule family. This allowed us to test on a schedule family with a large number of parameters. We trained with two independent sets of 100 seeds, and used the median training error of one set as a reference. We then compared this reference to medians computed using subsets from the second set of seeds — 1, 3, 10, and 100 seeds (Figure 9). We found that there was a high amount of variation between the reference and the 1 and 3 seed measurements, particularly at low error. This was an issue because our protocol requires us to use the small-number-of-seed experiments to suggest candidates for further measurement. We deemed that the 10 seed measurements displayed small enough variability to be sufficient for our purposes. In addition, the low discrepancy between the 100 seed measurements motivated our use of 100 seeds for the evaluation step.

We further tested our selection procedure by ranking the top 100 schedules (by lowest error) for each metric — the 100 seed reference, as well as the 4 measurements from the other 100 seed set. Then we took the union of the top 100 reference seeds with each of the top 100 measurements individually and plotted their errors in each measurement (Figure 10). The resulting measurement allowed us to understand the distribution of top performing schedules in each measurement. We defined a false negative rate by asking how many seeds ranked in the top 100 of the reference were *not* in the top 100 of the second measurement. Comparing the two independent 100 seed measurements gave us a baseline false negative rate of 0.17; this represents an empirical measurement of the baseline level of uncertainty given that our final evaluations are over 100 seeds. We found that with 1 measurement seed the false negative rate was 0.41. We saw that using 10 seeds for the measurement gave a false negative rate of 0.25, which we deemed close enough to the noise floor. Looking at the individual datapoints, we also see that the very best schedules in both measurements are in the top 100 for each measure, suggesting that our selection procedure (for this workload and schedule family) is well tuned to balance computational cost with measurement precision.

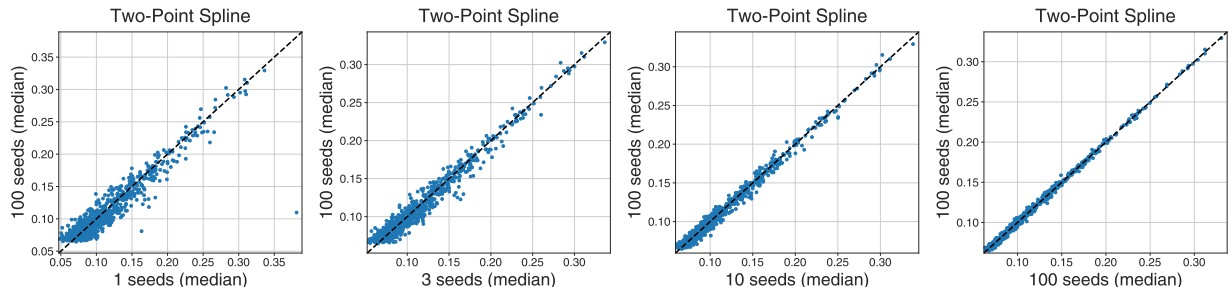

Figure 9: Median CIFAR-10 training error under the TWO-POINT SPLINE schedule family at four seed counts. Each plot compares the median from 1, 3, 10, or 100 seeds on the horizontal axis with the reference median from 100 seeds on the vertical axis. A single seed produces wide scatter. Three seeds reduce the spread but still drift. Ten seeds lie close to the diagonal and roughly match the reference, showing that ten seeds give a stable estimate.

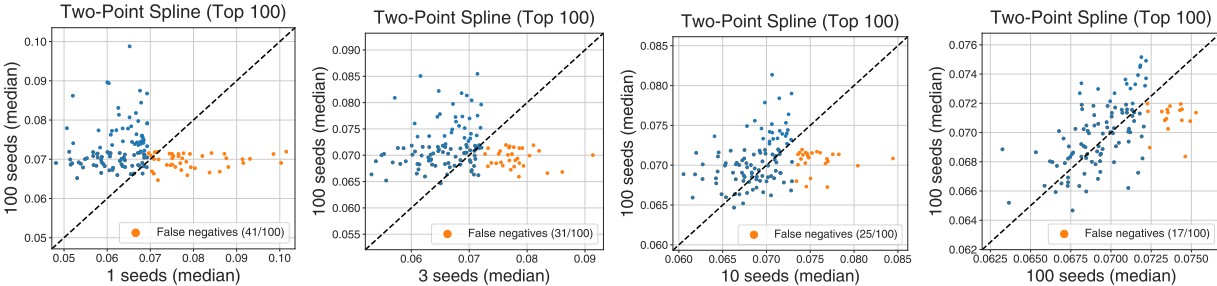

Figure 10: To test our selection procedure, we selected the top 100 schedules according to the reference measurement (100 seed median, $y$-axis) and smaller seed data ($x$-axis, various number of seeds). When a single seed was used for measurement, we found that 41 of the 100 schedules with lowest error according to the reference were not in the top 100 of the single-seed measurement (left). In comparison, using a 100 seed measurement, there were 17 out of the top 100 references which were not in the top 100 of the new measurement. We found that using 10 seeds for the measurement maintained similar false negative rates to the 100 seed measurement, and captures most of the best performing seeds. This motivated our choice of 10 seeds for the initial phase of the measurement protocol.

# C  Additional results

## C.1  Effects of non-zero final decay value

In all of our schedules except SMOOTH NON-MONOTONIC, we set the final learning rate to 0 at the end of training. Some training suggestions use a non-zero final learning rate, such as Hoffmann et al. (2022) which uses a cosine learning rate schedule with decay to 10% of the peak value. In this section, we explore the effects of allowing non-zero decay adding non-zero decay, and show that it doesn't change our results much.

We defined two additional schedules: COS-Y and TPS-Y which added a free parameter representing the fraction of the peak LR to decay to the COSINE and TWO-POINT SPLINE schedules respectively. We then performed the same search procedure as for the other schedule families on both workloads.

We found that a nonzero final base learning rate improved COS-STD on CIFAR-10 (Figure 11, left), and closed most (but not all) of the gap with COS-GEN. For CIFAR-10, COS-Y had a similar performance to COS-STD. This suggests that a tunable final base learning rate does indeed improve performance over COS-STD, but at least on our workloads tuning the power of the cosine is a better use of an additional parameter than tuning the final base learning rate.

In contrast, for the TWO-POINT SPLINE schedule we found that adding a tunable final base learning rate did not significantly change performance. The top 5 TPS-Y schedule all had a final decay value near 0, and consequently did not show significant differences in performance from TPS. This suggests that TWO-POINT SPLINE, and most likely, the other schedules with similar final optimal shapes do not benefit from non-zero final learning rate.

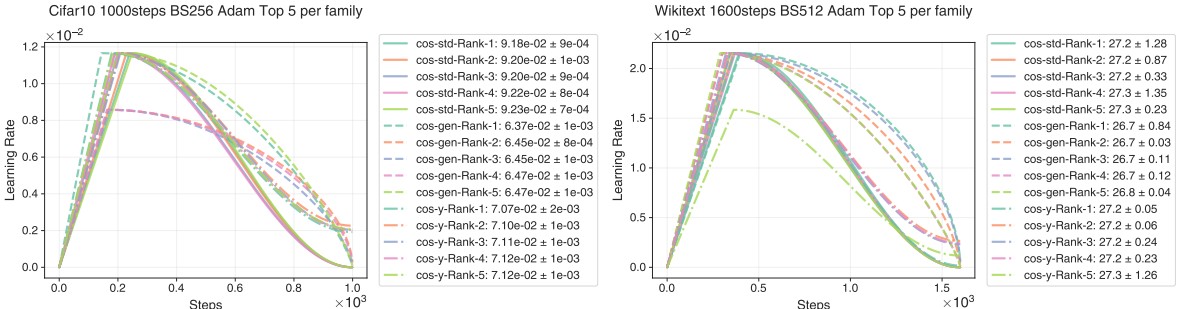

Figure 11: Top-5 schedules from search for COS-STD, COS-GEN, and COS-Y families. For CIFAR-10 non-zero final learning rate closes most, but not all of the gap between COS-STD and COS-GEN (left). For WIKITEXT-103, optimal $y$ value is close to 0 and there is no gain.

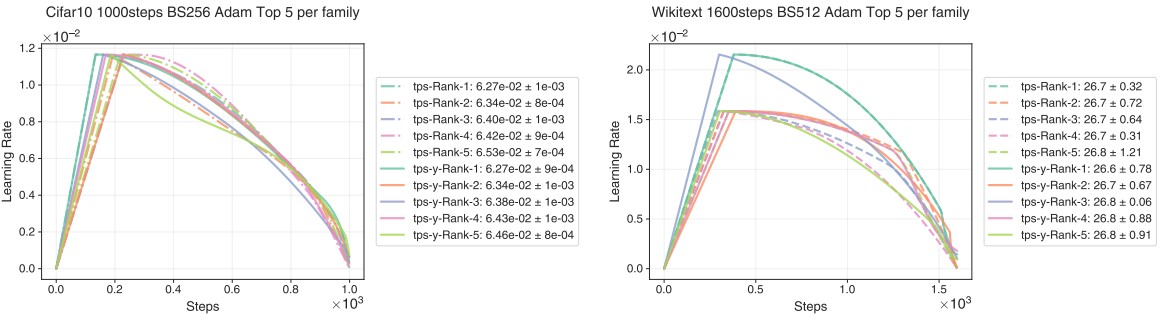

Figure 12: Top-5 schedules from search for TPS and TPS-Y families. Optimal final learning rate values are close to 0 and there are no appreciable differences between the families.

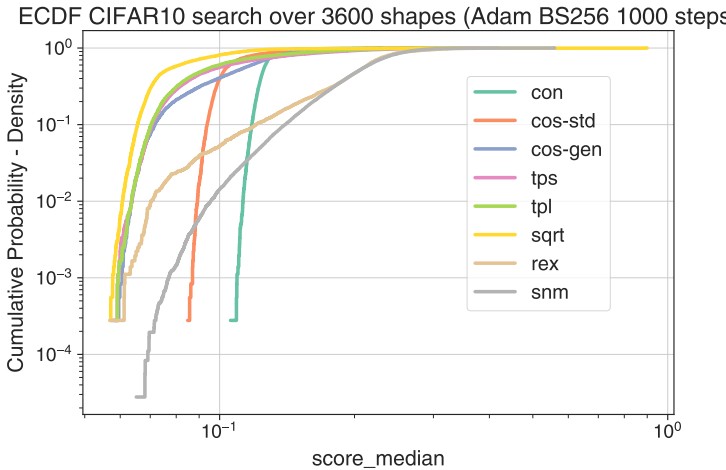

Figure 13: ECDF (Empirical cumulative distribution) of the best score found by random search in each schedule family. The data is the same as Figure 20, left panel, except the SMOOTH NON-MONOTONIC family was searched with 36000 families (10× more than the rest). The extra samples find schedules with better error than the original experiments, but SMOOTH NON-MONOTONIC lags behind the other families (higher error). This supports our claim that the SMOOTH NON-MONOTONIC family was not sufficiently searched/optimized relative to the other families.

## C.2 Smooth Non-Monotonic with increased samples

To confirm the finding that the SMOOTH NON-MONOTONIC family is not efficiently searched compared to the other families, we performed an additional search over 36000 shapes for SMOOTH NON-MONOTONIC in the CIFAR-10 workload. Plotting the ECDFs, we found that the additional samples did improve the best schedule found, but only by a small amount, leaving a gap with the other schedule families (Figure 13). This suggests that there is room for improvement in the SMOOTH NON-MONOTONIC family (as suspected since it is more flexible than all our other families), but that our search procedure is unable to efficiently probe the search space and requires improvement either of the search space geometry/search distribution, or needs new techniques altogether.

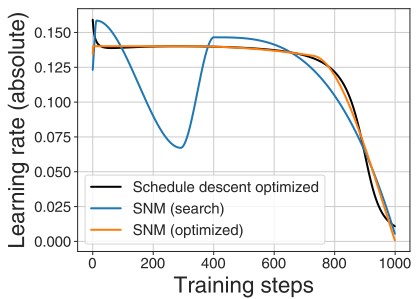 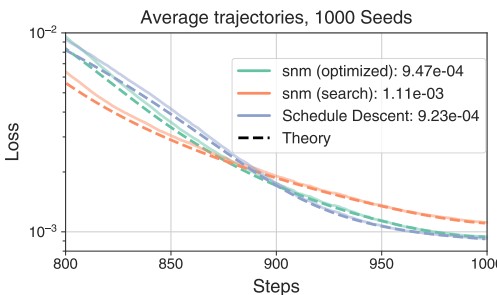

Figure 14: Best Smooth Non-Monotonic family member does not match optimal schedule (left, blue). We can numerically solve for the best fit curve in the family (left, orange). This obtains very similar performance to the theoretical optimal schedule (right).

## D  SNM comparison with optimal schedule (linear regression)

For the linear regression workload our random search methodology retrieves schedules which perform well overall and have some of the features of the true optimal schedule, but does not actually obtain the best schedules in each family. This can be seen most clearly with the Smooth Non-Monotonic family. We found the Smooth Non-Monotonic family member that is the closest to the ground truth via numerical optimization (Figure 14, left). The resulting schedule is worse than the true optimal schedule by $2 \cdot 10^{-5}$ only—much less than the difference of $8.5 \cdot 10^{-5}$ found from the random search (Figure 14, right).

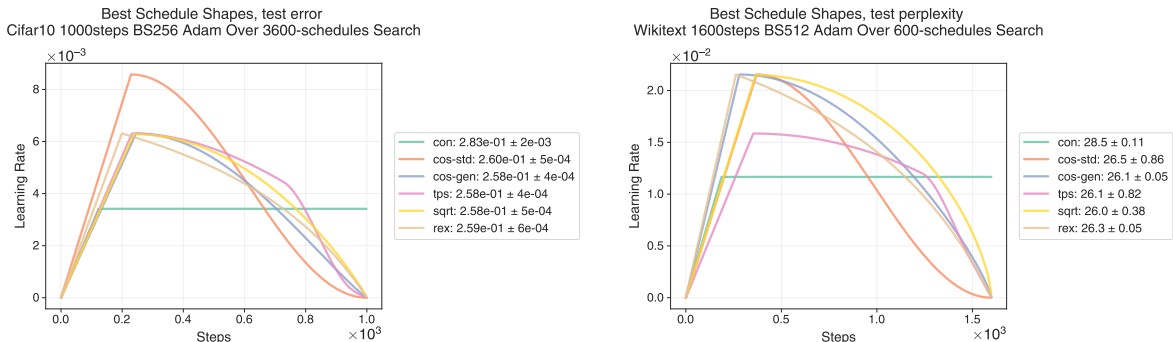

Figure 15: Near optimal learning rate schedules for CIFAR-10 (left) and WIKITEXT-103 (right), selected on best median test error (CIFAR-10) and test perplexity (WIKITEXT-103). Similar trends to selecting on training metrics, with some quantitative differences. CIFAR-10 favors training curves with smaller absolute base learning rate when selecting on test error vs train error. SNM and TPL were omitted for clarity.

## E  Near-optimal schedules selected on test metrics

We found similar results when selecting schedules using test metrics (error and perplexity for CIFAR-10 and WIKITEXT-103 respectively), with somewhat increased variation in schedule shapes and smaller base learning rates in the case of CIFAR-10 (Figure 15).

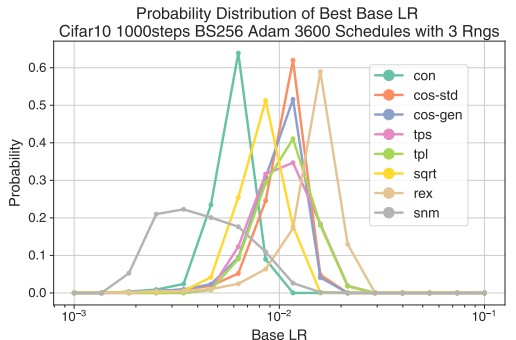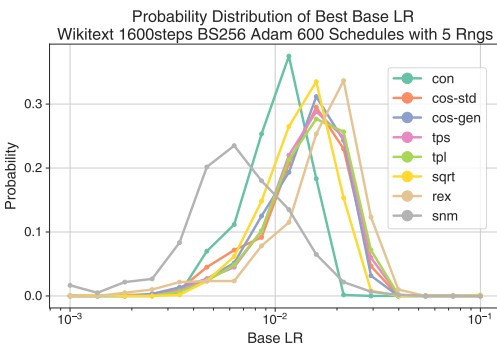

Figure 16: Distribution of best base learning rates for shape search points on CIFAR-10 (left) and WIKITEXT-103 (right). Each curve represents a different schedule family. Support for the distributions is mostly contained inside the search grid, which suggests that the base learning rate range was well-chosen. Optimal base learning rate distributions are similar across families, which the exception of CONSTANT and SMOOTH NON-MONOTONIC which are significantly smaller than the others.

## F   Validating the "near-optimal" nature of the search

In order for the analysis of the previous section to be meaningful, we must understand: to what extent is our search procedure really obtaining near optimal schedules? In this section we provide evidence that most of our families are indeed searched well, and that we expect our basic conclusions to hold even with a better search procedure.

We noted that the base learning rate is the most important factor in predicting the performance of a learning rate schedule (Figure 3); this is what motivated our decision to optimize the base learning rate for each family individually. The distribution of optimal learning rates suggests that the range of our grid search was chosen appropriately for each schedule shape (Figure 16).

Another relevant question is: how meaningful are the schedule shapes themselves? If the random search was undersampled, then the top shapes per family may in fact be highly variable. For CIFAR-10, we found that the top shapes were in fact relatively stable; this was true across families (Figure 4, left), but also for the top 3 shapes within a family (Figure 17, left, for COSINE and TWO-POINT SPLINE). For the WIKITEXT-103 workload there is more variability between the best schedule per family (Figure 4, right), though there is broad consistency; there is more consistency within family (Figure 17, right, for COSINE and TWO-POINT SPLINE). The exception in both workloads is the SMOOTH NON-MONOTONIC family, which both deviates from the other families, as well as has high within-family deviation even for the top schedules (Figure 17).

The poor optimization of the SMOOTH NON-MONOTONIC family can be seen directly by performing line-searches of schedules in different coordinate directions. For a near-optimal schedule, perturbing any individual parameter would not give improvements to the training metrics; for non-optimal schedules, at least one perturbation direction would give clear improvements. We carried out such a search on the best SMOOTH NON-MONOTONIC and TWO-POINT SPLINE schedules for both our workloads. We varied one schedule parameter at a time, reoptimized base learning rate, using 100 seeds for each coordinate searched. We found that for the SMOOTH NON-MONOTONIC schedule, for CIFAR-10 there were two coordinates – $\Delta x_2$ and $y_2$—which could be varied to significantly improve the final train error (Figure 18 (a)). For WIKITEXT-103, $y_2$ was also likely suboptimal, and $y_{start}$ may be as well; detailed analysis is made more difficult by the larger confidence intervals and smaller improvements available for WIKITEXT-103 (Figure 18 (b)). In contrast for the best TWO-POINT SPLINE schedule, only the coordinate $x_0$ was measurable suboptimal, and even there the best value found was within the 95% CI of the original point (Figure 19). This provides a more quantitative insight into how "near-optimal" our best TWO-POINT SPLINE schedule is, and show directly that the SMOOTH NON-MONOTONIC schedule is far less optimized than TWO-POINT SPLINE and other schedules.

As we discussed in Section 4.1 in the context of the linear regression workload, the SMOOTH NON-MONOTONIC family is not due to some fundamental limitation of its representation, but by the difficulty of

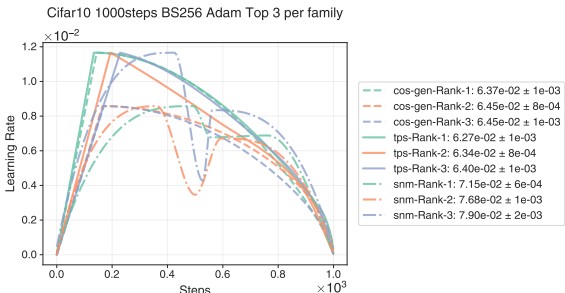 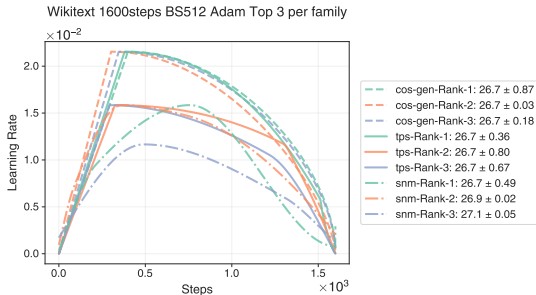

Figure 17: Top 3 schedules for CIFAR-10 and WIKITEXT-103 for COSINE, TWO-POINT SPLINE, and SMOOTH NON-MONOTONIC families. For COSINE, schedules are very similar within workload which suggests that COSINE was searched well. For CIFAR-10, all TWO-POINT SPLINE schedules are similar to each other, while for WIKITEXT-103 all but one TWO-POINT SPLINE schedules have similar warmup, base learning rate, and decay. This provides evidence that TWO-POINT SPLINE is also generally searched well, but that the WIKITEXT-103 workload admits a wider variety of learning rate schedules with similar final training metrics. SMOOTH NON-MONOTONIC schedules show high variability, suggesting they are not well-searched.

optimizing a higher dimensional family using random search alone. We can see this by plotting the empirical cumulative distribution function (ECDF) of the training metrics (Figure 20). The ECDF plots the selection score on the $x$-axis and the normalized rank on the $y$-axis; in other words, it is the inverse of the function $s(r)$ going from normalized rank to the score at that rank in a dataset.

For random search, given a true CDF function $C(s)$ the expected best score of a search over $k$ random samples is given by $s^*$ such that $C(s^*) = k$. Therefore the ECDFs give us a way to estimate the sample efficiency of random search; ECDFs for efficient methods will reach lower score values for equivalent normalized ranks (efficient methods generate ECDFs that reach the lower left of the plots). We can see that the CONSTANT and COSINE ECDFs converge to a point to the right of the other families; SMOOTH NON-MONOTONIC in contrast has an ECDF curve that is still traveling to the left (non-zero slope until the final few ranks), but does not reach the other schedules. This gives us additional evidence that the sample efficiency of the search is a problem. Indeed, increasing the number of samples for SMOOTH NON-MONOTONIC specifically by a factor of 10 still does not close the gap (Appendix C.2).

The difficulty of optimizing SMOOTH NON-MONOTONIC schedules with random search is due to the fact that warmup from (near) zero and decay to (near) zero learning rate is a key feature of the best schedules. With our random search procedure, the prior corresponding to these features is small for the SMOOTH NON-MONOTONIC schedule. A better search procedure (either random search with a better prior, or some adaptive search method) would likely greatly improve our ability to find near-optimal SMOOTH NON-MONOTONIC schedules.

Overall our results suggest that all our schedules save the SMOOTH NON-MONOTONIC can be reasonably considered to be "near-optimal". This suggests that for small workloads, our random search procedure can be used to reasonably optimize schedule families which naturally incorporate warmup and decay.

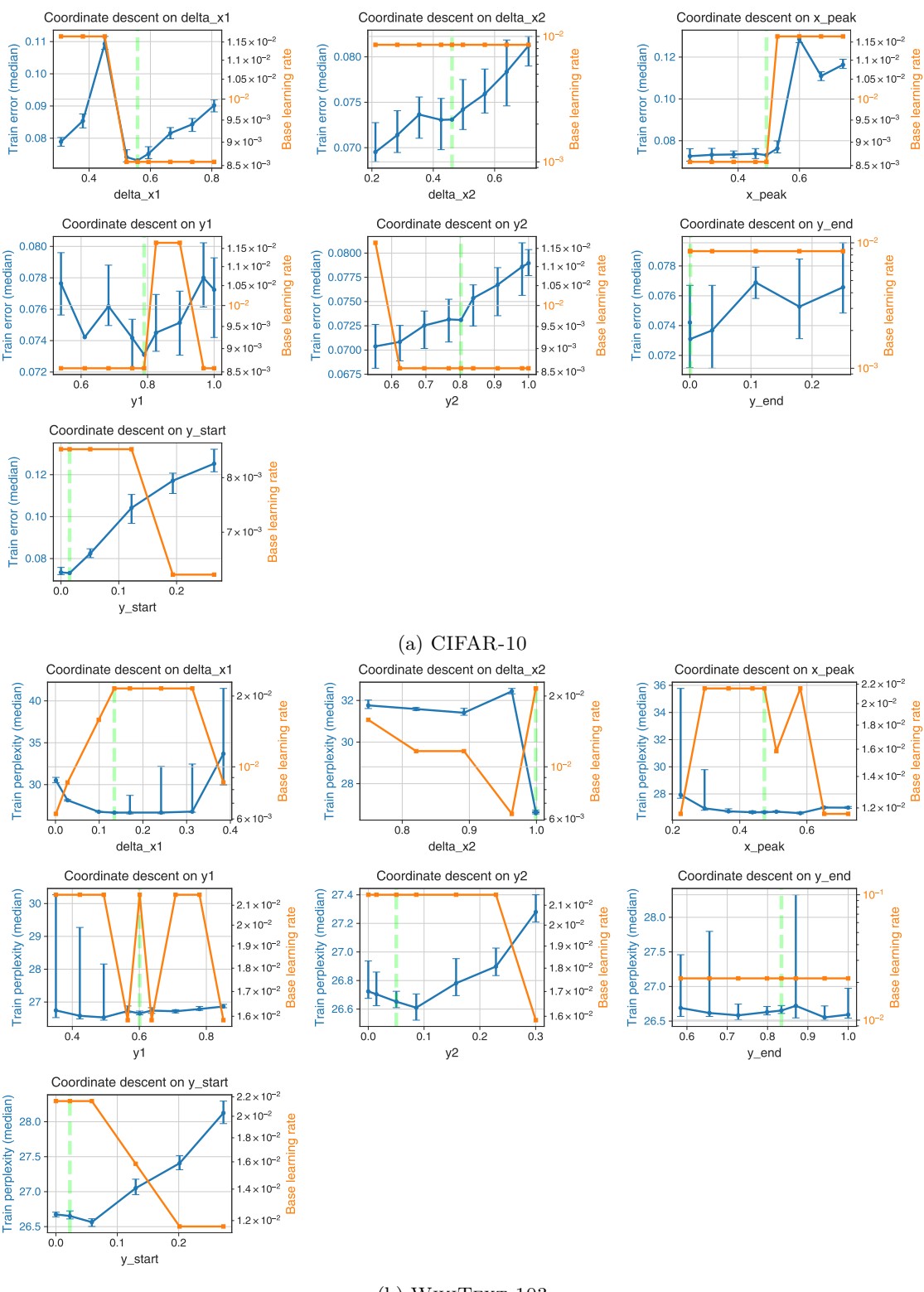

(a) CIFAR-10

(b) WikiText-103

Figure 18: Coordinate-wise linesearch from optimal Smooth Non-Monotonic schedule shape for CIFAR-10 (top 3 rows) and WikiText-103 (bottom 3 rows). For each coordinate, we vary the coordinate and remeasure the appropriate training metric (blue, 95% CI plotted), with a new base learning rate optimization (best base LR in orange). Green dashed line marks the original value of the coordinate from the search. Varying some coordinates, particularly $y_2$, can significantly improve training metrics suggesting that Smooth Non-Monotonic is not well-optimized.

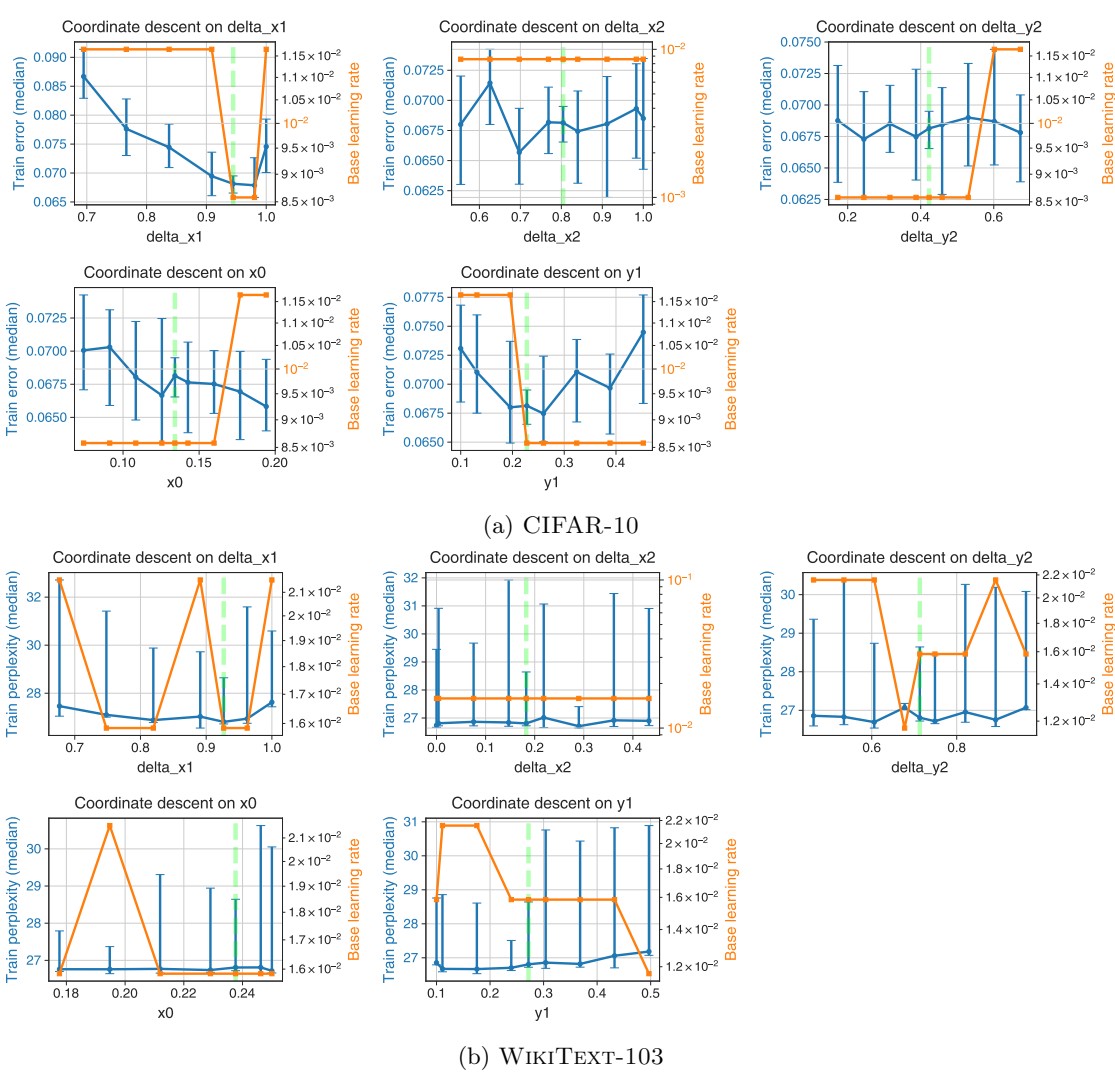

(a) CIFAR-10

(b) WikiText-103

Figure 19: Coordinatewise linesearch for Two-Point Spline schedule. Methodology and labeling are the same as Figure 18. Gains from linesearch are generally smaller than measurement uncertainty, suggesting Two-Point Spline is well-optimized at the resolution of our experiments — unlike Smooth Non-Monotonic.

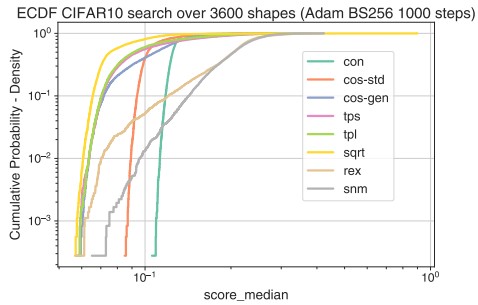 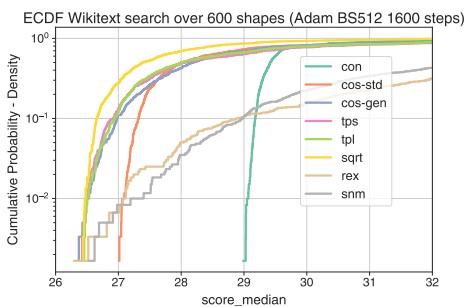

Figure 20: ECDF (Empirical cumulative distribution functions) of the score reached by random search within each learning rate schedule family. The horizontal axis reports the median training error for CIFAR-10 or the median training perplexity for WIKITEXT-103, where lower values mark better learning. The vertical axis shows the chance of reaching or beating each score. On CIFAR-10, the constant schedule and the cosine schedule with exponent 1.0 reduce error with high probability yet stall near 0.05. SQUARE-ROOT DECAY schedules move the curve left most sharply. TWO-POINT SPLINE, TWO-POINT LINEAR and GENERALIZED COSINE trace a similar path. GENERALIZED REX and SMOOTH NON-MONOTONIC attain lower scores at lower CDF probability (alternatively, after more samples). SMOOTH NON-MONOTONIC in particular lags because its large parameter space slows the search.

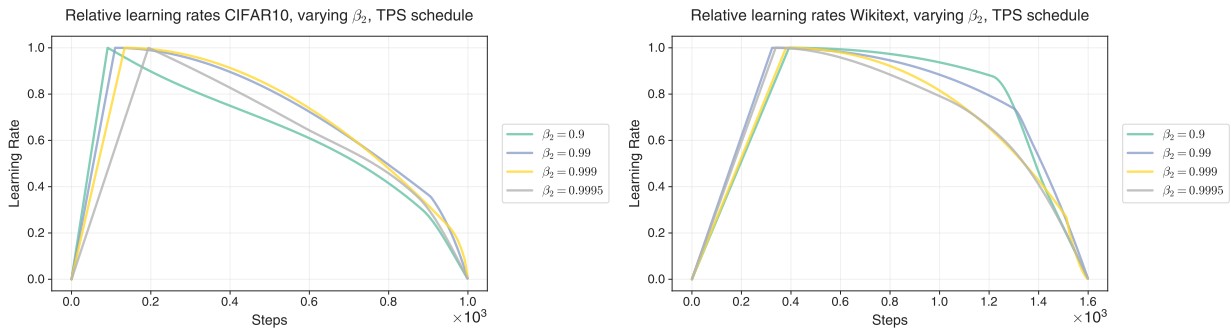

Figure 21: Optimal schedule shape varying $\beta_2$. Relative learning-rate schedules that minimise loss after tuning $\beta_2$. Left panel shows CIFAR-10; right panel shows WIKITEXT-103. On CIFAR-10, larger $\beta_2$ values extend the warmup, whereas $\beta_2$ shows no clear link to schedule shape on WIKITEXT-103.

## G    Effect of $\beta_2$ on optimal schedule shape

Figure 21 shows the optimal schedule shapes when varying $\beta_2$ for both workloads. On CIFAR-10, larger $\beta_2$ values appear to extend the warmup phase, whereas on WIKITEXT-103, $\beta_2$ shows no clear link to schedule shape. Our experiments do not provide convincing evidence for the value of $\beta_2$ having much effect on which schedule shapes perform well in any setting.

## H   Cross-evaluation tables for workload variations

This section contains the cross-evaluation tables for the workload variation experiments described in Section 4.3. Each table shows the training metric when using a schedule selected under one hyperparameter condition (rows) and evaluated under a different condition (columns).

### H.1   $\beta_1$ cross-evaluation

| $\beta_1$, Selection condition | $\beta_1$, Evaluation condition | | | |
| --- | --- | --- | --- | --- |
| | 0.8 | 0.9 | 0.95 | 0.975 |
| 0.8 | $0.07 \pm 0.002$ | $0.072 \pm 0.002$ | $0.089 \pm 0.001$ | $0.123 \pm 0.002$ |
| 0.9 | $0.068 \pm 0.002$ | $0.067 \pm 0.001$ | $0.083 \pm 0.001$ | $0.116 \pm 0.002$ |
| 0.95 | $0.066 \pm 0.001$ | $0.066 \pm 0.001$ | $0.082 \pm 0.001$ | $0.114 \pm 0.001$ |
| 0.975 | $0.068 \pm 0.004$ | $0.065 \pm 0.001$ | $0.081 \pm 0.001$ | $0.11 \pm 0.001$ |

Table 5: Median CIFAR-10 training error with one standard deviation for every pair of $\beta_1$ values used at schedule selection (rows) and evaluation (columns). A larger $\beta_1$ in the search phase lowers the error. The lowest error appears when the schedule found at $\beta_1 = 0.95$ runs with $\beta_1 = 0.8$. In most rows a lower $\beta_1$ at evaluation further reduces the error, which suggests that a drop in momentum after a high-momentum search can improve performance.

| $\beta_1$, Selection condition | $\beta_1$, Evaluation condition | | | |
| --- | --- | --- | --- | --- |
| | 0.8 | 0.9 | 0.95 | 0.975 |
| 0.8 | $28.8 \pm 0.96$ | $28.2 \pm 0.09$ | $33.04 \pm 0.06$ | $46.67 \pm 0.59$ |
| 0.9 | $37.58 \pm 1.91$ | $27.01 \pm 0.92$ | $27.11 \pm 0.02$ | $30.97 \pm 0.06$ |
| 0.95 | $39.27 \pm 1.85$ | $26.99 \pm 1.02$ | $26.91 \pm 0.04$ | $30.51 \pm 0.04$ |
| 0.975 | $40.61 \pm 4.09$ | $26.92 \pm 0.92$ | $26.91 \pm 0.08$ | $30.47 \pm 0.07$ |

Table 6: Median WIKITEXT-103 training perplexity and one standard deviation for every pair of $\beta_1$ values used in schedule selection (rows) and evaluation (columns). Rows list the $\beta_1$ that guides the search; columns give the $\beta_1$ used to run the schedule. A higher $\beta_1$ in the search stage lowers perplexity, except for selection at $\beta_1 = 0.975$, which yields larger values. The best result appears when schedules found at $\beta_1 = 0.95$ or 0.975 run with $\beta_1 = 0.95$. Evaluation at $\beta_1 = 0.8$ shows the widest spread, which points to high uncertainty for low-momentum runs.

### H.2   $\beta_2$ cross-evaluation

| $\beta_2$, Selection condition | $\beta_2$, Evaluation condition | | | |
| --- | --- | --- | --- | --- |
| | 0.9 | 0.99 | 0.999 | 0.9995 |
| 0.9 | $0.061 \pm 0.001$ | $0.062 \pm 0.001$ | $0.07 \pm 0.001$ | $0.072 \pm 0.001$ |
| 0.99 | $0.062 \pm 0.001$ | $0.062 \pm 0.001$ | $0.066 \pm 0.001$ | $0.068 \pm 0.001$ |
| 0.999 | $0.064 \pm 0.002$ | $0.064 \pm 0.001$ | $0.072 \pm 0.002$ | $0.07 \pm 0.002$ |
| 0.9995 | $0.058 \pm 0.001$ | $0.059 \pm 0.001$ | $0.066 \pm 0.001$ | $0.065 \pm 0.001$ |

Table 7: Median CIFAR-10 training error $\pm$ SD for every pair of $\beta_2$ values at schedule selection (rows) and evaluation (columns). Rows list the $\beta_2$ used to pick the schedule; columns give the value used to run it. Higher $\beta_2$ at selection tends to cut error, yet the lowest error comes from schedules tuned at $\beta_2 = 0.9995$ and run at $\beta_2 = 0.9$.

| $\beta_2$, Selection condition | $\beta_2$, Evaluation condition | | | |
|---|---|---|---|---|
| | 0.9 | 0.99 | 0.999 | 0.9995 |
| 0.9 | $26.23 \pm 0.02$ | $26.4 \pm 0.29$ | $26.87 \pm 6.03$ | $26.77 \pm 11.37$ |
| 0.99 | $26.32 \pm 0.01$ | $26.29 \pm 0.35$ | $26.59 \pm 0.85$ | $26.71 \pm 2.26$ |
| 0.999 | $26.32 \pm 0.01$ | $26.56 \pm 0.38$ | $26.74 \pm 1.42$ | $26.76 \pm 1.21$ |
| 0.9995 | $26.36 \pm 0.01$ | $26.42 \pm 0.27$ | $26.75 \pm 2.11$ | $26.74 \pm 2.77$ |

Table 8: Median WIKITEXT-103 perplexity $\pm$ SD for every pair of $\beta_2$ values used at schedule selection (rows) and evaluation (columns). A higher $\beta_2$ at selection lowers the median, yet the best result appears when a schedule tuned at $\beta_2 = 0.9995$ runs with $\beta_2 = 0.9$. Selection at $\beta_2 = 0.9$ shows the widest spread, confirming that low $\beta_2$ during tuning adds noise.

### H.3 Weight decay cross-evaluation

| $\lambda_{WD}$, Selection condition | $\lambda_{WD}$, Evaluation condition | | | |
|---|---|---|---|---|
| | 0.0 | 0.001 | 0.01 | 0.1 |
| 0.0 | $0.071 \pm 0.001$ | $0.127 \pm 0.003$ | $0.532 \pm 0.004$ | $0.9 \pm 0.0$ |
| 0.001 | $0.067 \pm 0.001$ | $0.123 \pm 0.002$ | $0.477 \pm 0.003$ | $0.9 \pm 0.0$ |
| 0.01 | $0.101 \pm 0.001$ | $0.149 \pm 0.001$ | $0.406 \pm 0.004$ | $0.9 \pm 0.002$ |
| 0.1 | $0.109 \pm 0.001$ | $0.164 \pm 0.001$ | $0.415 \pm 0.002$ | $0.839 \pm 0.004$ |

Table 9: Median CIFAR-10 training error $\pm$ SD for every pair of weight decay values used in schedule selection (rows) and later evaluation (columns). An evaluation decay of 0.01 or 0.1 lifts the error across all rows, while decay 0 or 0.001 yields the lowest values.

| $\lambda_{WD}$, Selection condition | $\lambda_{WD}$, Evaluation condition | | | |
|---|---|---|---|---|
| | 0.0 | 0.001 | 0.01 | 0.1 |
| 0.0 | $26.75 \pm 1.14$ | $29.2 \pm 0.78$ | $122.39 \pm 15.95$ | $5336.9 \pm 21.36$ |
| 0.001 | $27.08 \pm 0.72$ | $27.05 \pm 0.35$ | $101.69 \pm 12.31$ | $4169.88 \pm 43.03$ |
| 0.01 | $27.8 \pm 0.08$ | $27.61 \pm 0.06$ | $69.48 \pm 4.34$ | $443.55 \pm 21.18$ |
| 0.1 | $29.16 \pm 0.79$ | $30.93 \pm 3.52$ | $165.99 \pm 33.78$ | $392.74 \pm 10.19$ |

Table 10: Median WIKITEXT-103 training perplexity $\pm$ SD for every pair of weight-decay values used in schedule selection (rows) and later evaluation (columns). Lower decay during the search phase gives smaller medians. At evaluation, decay 0 or 0.001 holds perplexity down, whereas 0.01 or 0.1 drives it up.

# I  Training horizon analysis

For the WIKITEXT-103 workload, we varied the training horizon—the total number of training steps $T$—for the GENERALIZED COSINE and TWO-POINT SPLINE schedules. The base learning rates were similar in all settings, corresponding to two adjacent values of our search grid with no clear pattern. Plotting the schedule shapes against the fraction of training steps reveals a trend towards gentler decay for larger number of training steps (Figure 22). We also see that warmup fraction seems stable across horizons, which suggests that setting warmup fraction may be a more robust strategy than searching over/fixing the number of warmup steps. Training horizon experiments are much less interesting on CIFAR-10 because training for even a bit longer than we did in our experiments quickly saturates the training error at zero errors, making it hard to stay in the optimization-limited regime or learn very much.

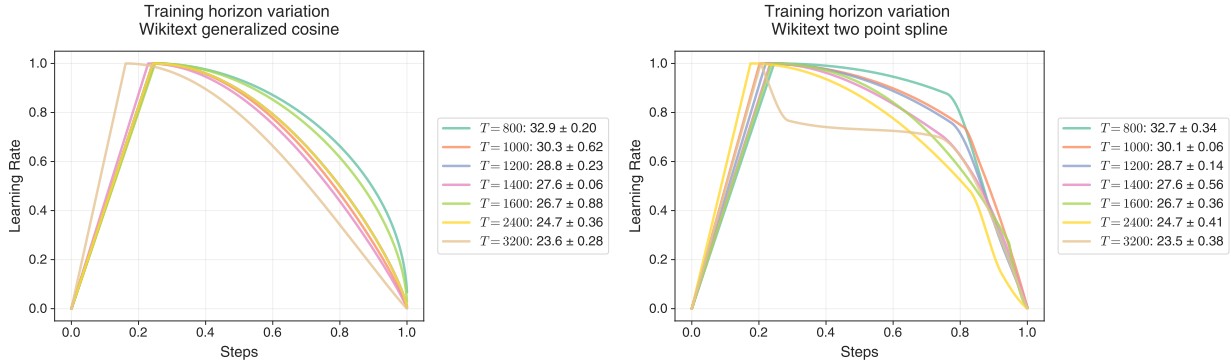

Figure 22: Optimal learning rate shapes for GENERALIZED COSINE (left) and TWO-POINT SPLINE (right) for WIKITEXT-103 workload trained with varying training horizon. Performance improves with more steps, which favors a more steady decay. Warmup fraction remains consistent at this scale.

