# OpenReview forum: "What do near-optimal learning rate schedules look like?"
_TMLR — Accepted by TMLR_

### Review · Reviewer_WYWc · 2026-04-08

**Summary Of Contributions:**

The paper studies the learning rate schedules. The paper develops a methodology to search the schedule families on computationally-inexpensive neural network training workloads to find near-optimal shapes within each family. The paper proves the optimal schedule for linear regression trained using SGD, and uses it to benchmark the efficacy of our search procedure. The paper provides near-optimal schedules for the different families defined on CNN trained to classify images and a small transformer model.  The paper also shows the relationship between optimal schedule shape in the workloads and other hyperparameters.

**Audience:**

Yes

**Audience Explanation:**

I believe so. How to choose learning rate schedule is a very practically relevant topic in machine learning.  Even though there is no theory provided in the paper, the extensive numerical results might help the readers in some practical problems.

**Claims And Evidence:**

Yes

**Claims Explanation:**

Yes and no. The paper is an empirical paper without any supporting theory.  The optimal learning rate schedules are searched within given schedule families as presented in Table 1. Even though this quite a large number of schedule families that are studied, without theoretical guarantees, it is hard to claim the learning rate schedules are optimal or nearly optimal. In theory, there can be infinitely many learning rate schedules, and it should be an infinite-dimensional optimization problem. Even though  a large number of schedule families are studied empirically, it is not easy to claim they are optimal or nearly optimal without any theoretical guarantees.

**Requested Changes:**

(1) The authors should mention the limitations of the work, especially the lack of any supporting theory.  This can for example be mentioned as a future research direction as well.

(2) In Appendix A.2., you use the high-dimension limit when $D$ is large. Is this approximation good when $P$ is fixed or when $P$ and $D$ go to infinity simultaneously where $P$ is the dimension of the parameter vector $\theta$? More discussions should be added.

(3) For the linear regression discussion in Appendix A.1 and A.2, as the author(s) noticed that by combining equations (8) and (9), one can write the loss as an analytic function of the learning rate schedule. The author(s) then commented that even if you approximate equation (8) using an ODE, it might not be analytically possible to solve for the optimal learning rate schedule. I am wondering the following thing. If you can combine equations (8) and (9), without using ODE approximation, it is already a well-defined optimization problem. It should be worth trying to study the complexity in terms of theory to solve such an optimization problem.

(4) This paper reminds me of recent works on learning to optimize, except that this paper specializes to finding the optimal learning rate schedule. It will be helpful if the author(s) can discuss the relation of this work to the existing literature on learning to optimize.

---

> ### Author Response · Authors · 2026-05-05
>
> We thank the reviewer for the careful reading and the constructive suggestions, and in particular for the technical engagement with the linear regression analysis in Appendix A. Manuscript additions corresponding to this response are highlighted in red.
>
> __[W1 / R1-(1)] On theoretical guarantees and limitations__
>
> We agree that the paper is empirical and does not provide formal optimality guarantees. We would like to clarify what "near-optimal" means in our setting and welcome the reviewer's feedback on how to describe this more precisely.
>
> For a fixed number of steps T, the optimal learning rate schedule is a T-dimensional vector. In our deep learning workloads T is on the order of $10^{3}$, and direct numerical optimization over this vector for each random seed and hyperparameter configuration is not tractable. We therefore restrict the search to parameterized families and use random search followed by re-evaluation with more seeds. We do not claim that the resulting schedules are globally optimal, only that they are the best we can find within each family using a finite search budget. We attempted to better quantify the degree to which our searches were optimal by performing coordinate-wise perturbations as well as looking at the tails of the CDFs of the validation metrics (Appendix F), both of which suggested that many of the families were searched well.
>
> In the linear regression setting we can do better, because the high-dimensional approximation gives us an analytic loss as a function of the T-dimensional schedule vector. This allowed us to implement Schedule Descent (Appendix A.3), which directly optimizes the schedule vector and gives us a numerical ground-truth optimum that we can use to validate the random search. Figure 2 and Figure 14 show that random search recovers shapes close to the ground truth, and that the gap can be closed further by more careful optimization within a family.
>
> We have added a clarification at the end of Section 5.2 making the above explicit — that "near-optimal" refers to the best schedule we can find within a parameterized family using a finite search budget, validated for each family in Appendix F — and we have explicitly noted theoretical analysis of near-optimality as a future research direction in the same paragraph. If the reviewer has suggestions for how to describe the sense in which our schedules are near-optimal more clearly, we would be glad to incorporate them.
>
>
> __[R1-(2)] Validity of the high-dimensional approximation__
>
> Equation (8) is derived in the regime where the parameter dimension $P$ and the data dimension $D$ both go to infinity with their ratio fixed, and with $B/D$ held fixed. This is the same regime used by Lee et al. (2022) and Agarwala & Pennington (2024). We noticed while preparing this response that the original manuscript was not explicit about the joint scaling of $P$ and $D$, and we have corrected this with an added sentence in Appendix A.2.
> Appendix A.5 compares the theoretical prediction to empirical averages over 1000 seeds: agreement is good at early and late times, with some deviation at intermediate times when the dynamics are near or above the edge of stability.
> We note that the high-dimensional limit also makes the search for optimal schedules numerically tractable: it gives us a closed-form loss as a function of the T-dimensional schedule vector, which we exploit in Schedule Descent.

---

> > ### Comment · Reviewer_WYWc · 2026-05-31
> > **response**
> >
> > Thanks for your response. My main concern was a lack of theory and the not-so-clear meaning of near-optimal. I think the author(s) have done a good job to explain what they mean by near-optimal, and they also explained the high-dimensional approximation in more detail.

---

> ### Author Response · Authors · 2026-05-05
>
> __[R1-(3)] On directly studying the optimization problem from Eqs. (8) and (9)__
>
> The reviewer is correct, and Schedule Descent (Appendix A.3) is the direct implementation of this idea. It uses Eqs. (8) and (9) without any ODE approximation, differentiating through the recurrence as an unrolled computation and performing gradient descent on the T-dimensional schedule vector. We have added a sentence in Appendix A.3 to make this connection explicit, stating that this is exactly the well-defined finite-dimensional optimization problem one obtains by combining the loss recurrence and the loss-from-state equation without any ODE approximation, by unrolling the recurrence for T steps.
>
> The reviewer also raises the good point that the resulting optimization problem is itself worth studying theoretically. We do not pursue this in the present paper, but we agree it is a natural follow-up; the sentence we added to Appendix A.3 makes this explicit, noting that the objective is non-convex in the schedule vector and that a detailed theoretical analysis of its complexity is a natural follow-up. This non-convexity is consistent with two empirical observations from our Schedule Descent runs: (i) the initialization of the schedule affects which optimum we converge to, and (ii) when a candidate update would cause training to diverge — at which point the gradient with respect to the schedule carries no useful signal — vanilla gradient descent cannot recover. To handle this case we fall back to a simple safeguard that uniformly shrinks all learning rates by a constant factor of 0.3 (Appendix A.3).
>
>
> __[R1-(4)] Connection to learning-to-optimize__
>
> We agree this connection should be made. We have added a brief discussion in Section 2 (Related Work) noting that our work is related to the learning-to-optimize literature (Andrychowicz et al., 2016; Wichrowska et al., 2017; Metz et al., 2019). The distinction we draw is that L2O learns optimizer update rules through meta-learning, often producing schedules only implicitly, while we directly characterize near-optimal schedule shapes within parameterized families. The two approaches are complementary: empirical characterization of near-optimal shapes gives a target that learned optimizers can be compared against.
>
>
> We thank the reviewer again for the suggestions, which have led to clearer positioning of the paper and meaningful additions to the related-work and limitations sections.

---

### Review · Reviewer_Ub44 · 2026-04-21

**Summary Of Contributions:**

This paper investigates the effect of the shape of learning rate schedules on training performance via a systematic suite of experiments on a range of learning rate schedules. The experiment setup consists of training on a linear regression task, CIFAR-10, and an 8M transformer on  Wikitext-103. The authors conclude that it is generally important to have a warmup and monotonic decay stage and to set the base learning rate accurately.

**Audience:**

Yes

**Audience Explanation:**

Learning rate scheduling is highly relevant to almost all areas of machine learning.

**Claims And Evidence:**

Yes

**Claims Explanation:**

The experimental setup is outlined in detail in Section 3 and the referenced appendices, and the results are discussed in Section 4.

**Requested Changes:**

While the results of this work are undeniably important, I have major reservations regarding how effectively and how accurately it addresses its stated goal of finding near-optimal learning rate schedules for deep neural network training. This is broadly due to two reasons:
- The scale of the experiments on CIFAR-10 and Wikitext-103 is *far* too small. The authors acknowledge that these "small, computationally-inexpensive workloads" were chosen so that "multiple training runs could fit on a single accelerator", but this does not negate the fact that these experiments were effectively conducted on toy examples in the context of modern deep learning. Consequently, from the results presented in this paper alone, I find it unconvincing that the authors' conclusions would also hold for large-scale models.
- Several important design choices in optimizers and learning rate schedules were not studied, including
  - schedules defined without a training horizon. The authors acknowledge this omission but do not provide any justification
  - nonlinear warmup. The experimental setup uses linear warmup for (almost) all shapes, without justification or exploration of other possibilities
  - the selection of shapes. To me, it seems like smooth nonmonotonic is by far the most general and most expressive of the shapes listed in Table 1, and yet the authors provide the fewest insights on this particular shape. This raises the question of whether the search over shapes was comprehensive enough to claim to have found a near-optimal shape. More generally, the approximation of $\theta^\star$ described in the two paragraphs below (2) seems highly heuristic, and there are no formal guarantees on the quality of the approximation
  - optimizers and techniques commonly used in modern deep learning, e.g. Lion [CLH+23], Muon [JJB+24], cautious weight decay [CLL+26].

In addition,
- (minor) typo in the last paragraph before Section 4.3: "which suggested that Two-Point Spline is well-optimized but Smooth Non-Monotonic is now"; the last word should be "not".
- (minor) in the paragraph title near the end of page 9, ensure that AdamW has the correct case.

# References
[CLH+23] Xiangning Chen, Chen Liang, Da Huang, Esteban Real, Kaiyuan Wang, Yao Liu, Hieu Pham, Xuanyi Dong, Thang Luong, Cho-Jui Hsieh, Yifeng Lu, Quoc V. Le. Symbolic Discovery of Optimization Algorithms. NeurIPS 2023.

[JJB+24] Keller Jordan, Yuchen Jin, Vlado Boza, Jiacheng You, Franz Cesista, Laker Newhouse, Jeremy Bernstein. Muon: An optimizer for hidden layers in neural networks. 2024.

[CLL+26] Lizhang Chen, Jonathan Li, Kaizhao Liang, Baiyu Su, Cong Xie, Nuo Wang Pierse, Chen Liang, Ni Lao, Qiang Liu. Cautious Weight Decay. ICLR 2026.

---

> ### Author Response · Authors · 2026-05-05
>
> We thank the reviewer for the careful reading and the detailed feedback. We are encouraged that the reviewer found our results "undeniably important" and answered Yes to both TMLR criteria. Before addressing each point, we want to clarify the scope of the paper, since several of the reviewer's concerns address claims we do not actually make. Manuscript additions corresponding to this response are highlighted in red.
>
> The paper's contribution is a search methodology that decouples schedule shape from base learning rate, together with an empirical characterization of near-optimal shapes within each of eight defined schedule families on three controlled workloads. We also study how the optimal shape depends on other AdamW hyperparameters. We do not claim that our findings generalize directly to LLM-scale training, that we have identified globally optimal schedules across all possible families, or that our results extend without modification to all modern optimizers. Each of these would be a separate study.
>
>
> __[R1] On the scale of the experiments__
>
> The choice of small workloads is deliberate and necessary for the methodology. The conclusions in our paper rely on $10^{7}$ training runs for CIFAR-10 and $10^{6}$ runs for Wikitext-103, which is what allows us to distinguish between schedule shapes whose performance differs by small but statistically meaningful amounts. This compute envelope is incompatible with large-scale training.
>
> We respectfully disagree with the characterization of CIFAR-10 and Wikitext-103 as toy examples. Both are standard ML benchmarks, and prior work studying related questions about base learning rate behavior (Shallue et al., 2019; Goyal et al., 2017) has used workloads of comparable scale. We have added a scope clarification at the end of Section 5.2 making the within-family, finite-budget nature of our "near-optimal" claims explicit, and we have listed scaling our search to LLM-scale training horizons as a concrete future direction in the conclusion (Section 6); a different methodology would be required for that regime.
>
>
> __[R2-1] Schedules without a training horizon__
>
> We accept that the omission was under-justified in the original manuscript, and we provide the rationale here. Fixed-horizon schedules were chosen to enable fair cross-schedule comparison with a controlled step budget, since horizon-free schedules introduce an additional axis of variation that would confound the comparisons we wanted to make. We have added "exploring horizon-free families" as an explicit future direction in the conclusion (Section 6). We did not insert a separate methodological justification into Section 3 in this revision in order to keep the manuscript changes minimal, but we are happy to do so if the reviewer prefers.
>
>
> __[R2-2] Nonlinear warmup__
>
> Linear warmup is the near-universal choice in current practice, and our research question is about decay shape given a standard warmup rather than the warmup shape itself. Exploring nonlinear warmup is a reasonable extension, but it is a separate question from the one we study. We have added "exploring … nonlinear warmup" as a future direction in the conclusion (Section 6). As with the previous point, we did not add a separate justification paragraph to Section 3.1 in this round in order to keep the changes minimal, but can do so if the reviewer would like that promoted into the main text.
>
>
> __[R2-3] Selection of shapes and SNM__
>
> We claim near-optimality within each defined family, not across all possible schedule families. The paper is explicit that SNM specifically was not fully optimized by random search (Section 4.2, Appendix C.2, Appendix F). We treat SNM as a maximally flexible control family that bounds what additional flexibility can buy, and we validate near-optimality for the other families through coordinate-wise linesearches and ECDF analysis (Appendix F). For TPS in particular, the linesearches show that no individual coordinate can be improved beyond the measurement noise floor.
>
> The approximation procedure following Eq. (2) is a practical search-and-evaluation strategy rather than an algorithm with formal optimality guarantees, and we agree that this should be stated explicitly. We have added a clarification at the end of Section 5.2 stating that "near-optimal" refers to the best schedule we can find within a parameterized family using a finite search budget — validated for each family in Appendix F — and that a theoretical analysis of near-optimality remains an open direction.

---

> ### Author Response · Authors · 2026-05-05
>
> __[R2-4] Modern optimizers__
>
> We focused on SGD and AdamW, as they are well studied and performant on our chosen workloads (and, in the case of AdamW, on many other workloads as well). Due to the breadth of our study we chose not to compare across algorithms as well; the methodology itself is optimizer-agnostic and extends naturally to Lion, Muon, and cautious weight decay (up to choices of hyperparameter tuning). We have added this as a direction for future work in the conclusion (Section 6).
>
>
> __[Minor corrections] Typo__
>
> "now" → "not" corrected (Section 4.2). AdamW casing fixed (Section 4.3 paragraph heading).
>
>
>
> We hope these responses clarify the paper's scope and how we have strengthened the manuscript in revision. The natural extensions the reviewer identifies are valuable directions, and we share the view that they are worth pursuing. We maintain, however, that they constitute separate studies, and that the contributions of the present paper stand on their own within the scope we have defined.

---

> > ### Comment · Reviewer_Ub44 · 2026-05-22
> >
> > I thank the authors for the comments and revision.
> >
> > I apologize for any unintended implication in my original review that the authors were overclaiming their contributions. That being said, I maintain the belief that the ultimate goal of such a study is to identify optimal or near-optimal configurations for a wide range of settings and architectures, as suggested by the use of broad language such as "neural network training" and "most comprehensive results on near-optimal schedule shapes" in the abstract. I would suggest the authors to be as explicit as possible regarding the scope and, particularly, the limitations of the methodology and results throughout the paper.
> >
> > However, I am willing to accept that additional results pertaining to, e.g., scalability or modern optimizers remain an ideal contribution instead of a necessary one, or perhaps deferred to future work as the authors suggest.
> >
> > Regarding horizon-free schedules and nonlinear warmup, I strongly recommend the authors justify any omission of potential design choices in order to improve the clarity and quality of the work. It would also be good to have additional discussion on the optimizer choice, the selection of shapes, and other possibilities.
> >
> > I appreciate the added material, and I believe my concerns can be satisfactorily addressed in a revision. Overall, this paper is a solid contribution, and I lean towards acceptance.

---

### Review · Reviewer_YyAV · 2026-04-22

**Summary Of Contributions:**

This paper presents a systematic empirical investigation of near-optimal learning rate schedule shapes for neural network training. The authors develop a search methodology that decouples schedule shape from base learning rate, enabling fair comparisons across different schedule families. They evaluate eight parameterized schedule families (Constant, Cosine, Generalized Cosine, Square-Root Decay, Generalized REX, Two-Point Spline, Two-Point Linear, and Smooth Non-Monotonic) on three workloads: linear regression (as a test case with ground truth), CIFAR-10 image classification, and Wikitext-103 language modeling.

Overall, the paper makes a solid empirical contribution to understanding learning rate schedules, with careful methodology and interesting findings. The main limitations are the small-scale workloads and the admitted difficulty in optimizing the most flexible schedule family. However, the work provides a valuable foundation for future research on larger-scale problems and more efficient schedule search methods.

**Audience:**

Yes

**Audience Explanation:**

The paper's systematic empirical characterization of near-optimal learning rate schedule shapes addresses a fundamental and practically important question in deep learning optimization that will interest both practitioners and theorists in TMLR's audience.

**Claims And Evidence:**

Yes

**Claims Explanation:**

The claims are supported by accurate and clear evidence through ground-truth validation on linear regression, systematic noise characterization justifying the experimental design, multiple consistency checks, and rigorous cross-evaluation tables demonstrating hyperparameter dependencies, all presented with transparent acknowledgment of limitations.

**Requested Changes:**

1. Your workloads use 1000-1600 steps, but modern LLM training uses hundreds of thousands to millions of steps. Do you have evidence that warmup fractions (10-30% of training) scale? Intuitively, spending 30% of a 1M-step training on warmup seems excessive.

2. Your CIFAR-10 experiments use batch size 256. How does optimal schedule shape depend on batch size?

3. You note that linear regression optimal schedules have no warmup and sharp decay, while deep learning benefits from warmup and gentle decay. What property of non-convex optimization causes this difference?

---

> ### Author Response · Authors · 2026-05-05
>
> We thank the reviewer for the careful reading and the thoughtful questions. We are grateful that the reviewer found our claims well-supported and the findings of interest to TMLR readers. We address each question below. Manuscript additions corresponding to this response are highlighted in red.
>
>
> __[R1] On warmup fraction scaling to longer training horizons__
>
> The reviewer's intuition is well-founded. We do not claim that the warmup fractions observed in our experiments should be applied directly to million-step training. How warmup duration scales to large horizons is outside the scope of this work, which focuses on a regime where exhaustive search is feasible.
>
> Our manuscript does provide preliminary evidence within the range we explored. In Appendix I, we vary the Wikitext-103 training horizon from 800 to 3200 steps and find that the optimal warmup fraction remains stable while the decay becomes gentler with longer horizons. This suggests that warmup fraction may transfer better than absolute warmup steps, though whether this stability persists at LLM scale remains open. We have added scaling our search to LLM-scale training horizons as an explicit future direction in the conclusion (Section 6). We have not run experiments at substantially larger scales, since our search methodology (over $10^{7}$ training runs for CIFAR-10) is not compatible with that regime.
>
> __[R2] On batch size dependence__
>
> We did not study batch size dependence, and we agree it is an interesting question. Prior work has examined how the optimal base learning rate depends on batch size (Shallue et al., 2019; Goyal et al., 2017), but the interaction between batch size and schedule shape has not been systematically characterized.
>
> Our results in Section 4.3 give some indirect basis for expecting such an interaction. We found that $\beta_1$​ and weight decay can change the optimal shape, with weight decay having a particularly strong effect. Since batch size modulates the effective gradient noise, it is plausible that it would similarly affect the optimal shape. We have added "characterizing the dependence of optimal shape on batch size" as a future direction in the conclusion (Section 6). A full batch size sweep would require re-running the search at each batch size and would substantially expand the scope of the paper.
>
>
> __[R3] On the difference between linear regression and deep learning__
>
> This is one of the most interesting questions raised by our results. We do not have a complete answer, but we offer the following interpretation, which we have added as a paragraph in Section 5.2.
>
> The linear regression workload has fixed curvature: the Hessian eigenvalues are determined by the data covariance and do not change during training. In this setting, the optimal learning rate can be aggressive from the start, because the curvature will not evolve into a regime where this rate becomes unstable.
>
> Our analysis in Appendix A.4 shows that the optimal linear regression schedule operates near, and sometimes briefly above, the edge of stability, with sharp decay at the end used to converge the large eigenmodes.
>
> Deep learning workloads behave differently. Progressive sharpening (Cohen et al., 2022) and curvature dynamics near the edge of stability (Gilmer et al., 2021) mean that the largest Hessian eigenvalues can grow substantially during early training. Beginning at a large initial learning rate in this regime risks catastrophic instability before the curvature has settled. We hypothesize that warmup serves this function: it allows the curvature to reach a state in which the peak learning rate is tolerable. In contrast, at later times the edge of stability mechanism stabilizes a wider range of learning rates.
>
> These two effects taken together mean that 1. Deep learning workloads respond well to warmup and 2. With warmup there is a larger range of viable base learning rates.
>
> The preference for gentler decay in deep learning is consistent with this picture. The deep learning loss landscape has structure across many scales, and gentler decay may be needed to make balanced progress across directions throughout training. The linear regression problem with our chosen spectrum has a narrower range of relevant scales, and its late-time dynamics are dominated by a small number of slow modes that benefit from a final sharp decay.
>
> This is an interpretation rather than a demonstrated mechanism, and the relationship between curvature dynamics and optimal schedule shape remains an open problem. One contribution of our work is to sharpen this question empirically by establishing what near-optimal schedules look like in both settings.
>
>
>
> We hope these responses address the reviewer's questions. We view all three as natural extensions of our work rather than gaps in its present scope.

---

### Decision · Action_Editor_ncjp · 2026-06-01

**Recommendation:** Accept as is

**Audience:**

Yes

**Audience Explanation:**

Yes, those aiming to finetune models on relatively small datasets may find this insightful and better able to train their models.

**Claims And Evidence:**

Yes

**Claims Explanation:**

The authors have provided a reasonable study on learning rate schedules with good experimental results on two common datasets.  Questions raised by reviewers have already been incorporated into the current version of the paper.